# CLIN: A CONTINUALLY LEARNING LANGUAGE AGENT FOR RAPID TASK ADAPTATION AND GENERALIZATION

## ABSTRACT

Language agents have shown some ability to interact with an external environment, e.g., a virtual world such as ScienceWorld, to perform complex tasks, e.g., growing a plant, without the startup costs of reinforcement learning. However, despite their zero-shot capabilities, these agents to date do not continually improve over time, beyond performance refinement on a specific task. Here we present CLIN, the first language-based agent to achieve this, so that it continually improves over multiple trials, including when both the environment and task are varied, and without requiring parameter updates. Our approach is to use a persistent, dynamic, textual memory, centered on *causal abstractions* (rather than general "helpful hints"), that is regularly updated after each trial so that the agent gradually learns useful knowledge for new trials. In the ScienceWorld benchmark, CLIN is able to continually improve on repeated trials on the same task and environment, outperforming state-of-the-art reflective language agents like Reflexion by 23 absolute points. CLIN can also transfer its learning to new environments (or new tasks), improving its zero-shot performance by 4 points (13 for new tasks) and can further improve performance there through continual memory updates, enhancing performance by an additional 17 points (7 for new tasks). This suggests a new architecture for agents built on frozen models that can still continually and rapidly improve over time.

## 1 INTRODUCTION

Large language models (LLMs) have been increasingly used to interact with external environments (e.g., simulated worlds) as goal-driven agents (Reed et al., 2022). However, it has been challenging for these language agents to efficiently learn from trial-and-error as traditional reinforcement learning methods require extensive training samples and expensive model fine-tuning (Chen et al., 2021; Ammanabrolu et al., 2020). More recently, new techniques have appeared in which an agent reflects on its own past experience solving a task in a particular environment, and generates language-based insights to help it retry the task, e.g., Reflexion (Shinn et al., 2023). Such methods have the advantage of not requiring parameter updates (particularly useful given the growing popularity of frozen language models). However, the style of such insights plays a crucial role in performance, and not all insights improve generalization performance. For example, a specific insight such as "In the next trial, I will go to desk 1 and find the lamp" (Shinn et al., 2023) may have limited value (or even hurt) for a different environment or task.

Our goal is a system that will continually improve over time, both while attempting the same task in the same environment, and across different tasks and environments. Our approach builds on prior work on reflection in two ways: First, we conjecture that a specific *style* of insight will be useful, namely one that captures **causal abstractions** about agent's actions, e.g., "opening doors may be necessary for movement between rooms". Causal abstractions can potentially help the agent decide which action to take in the future, and can be viewed as a kind of action model learning (Arora et al., 2018), but placed in the modern context of language models. Second, we maintain these abstractions in a **continually evolving, dynamic memory**, which is regularly updated as the agent gains experience, allowing useful causal knowledge to persist (and unhelpful knowledge to be dropped) over time and between tasks and environments, as illustrated in Figure 1.

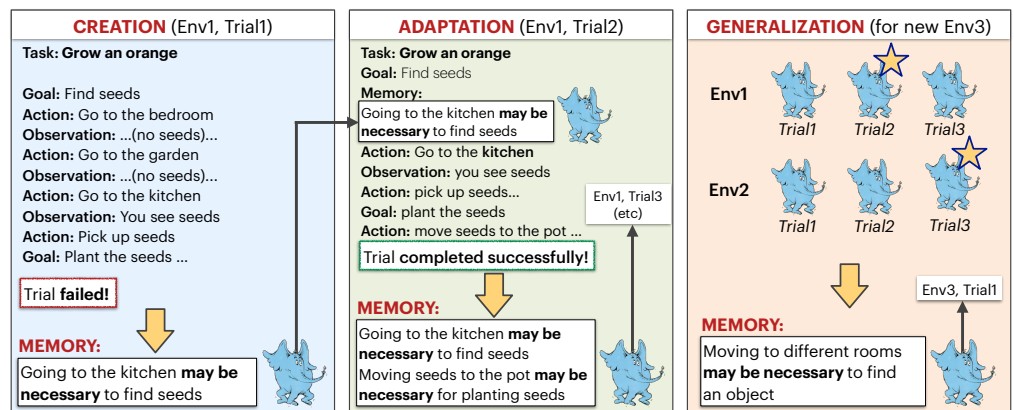

Figure 1: CLIN creates (Trial1) or adapts (Trial2+) a **memory of causal abstractions** to help in future trials by reflecting on the last trial and current memory. It does this using a suitably prompted LLM to generate the updated memory (Section 3.2). Here, reflecting on Trial1, CLIN notes in memory that going to the kitchen helped with finding seeds, enabling it to find the seeds faster in Trial2. From there, it also learns that moving the seeds to the pot helped plant the seeds. To further generalize across episodes (sequences of trials, right figure) for use in new environments, CLIN generates a summary ("meta-memory") of the best (starred) memories from each prior episode, here generating the generalization that moving to different rooms helps finding objects (Section 3.3)

We operationalize and evaluate this approach in a memory-augmented language agent called CLIN (**c**ontinual **l**earning from **in**teractions). CLIN is an agent that operates in ScienceWorld (Wang et al., 2022), a virtual, text-based environment in which an agent is tasked with science-based goals, e.g., boiling a liquid, growing a plant. We find that CLIN is able to rapidly learn about the environment and its action vocabulary and continually improve on repeated trials on the same task and environment, outperforming state-of-the-art (SOTA) reflective language agents like Reflexion by 23 points. CLIN can also transfer its learning to new environments (or tasks), improving its zero-shot performance by 4 (13 for new tasks) points and can further improve performance through continual memory updates, enhancing performance by an additional 17 (7 for new tasks) points. Our contributions are as follows:

- We describe and evaluate CLIN, an architecture for a novel nonparametric learning paradigm. We show using a dynamic, evolving memory over time, CLIN learns faster than the short-term "reflect, use, then discard" approach used in Reflexion and other memory-based agents and generalizes better to new tasks and new environments, achieving state-of-the-art.
- We show that memory of causal abstractions (or "action models") is effective at helping the agents learn over an extended period and for varying tasks and environments—first to apply in the modern context of language-based agents.
- Overall, this work suggests that a dynamic memory, centered around causal knowledge, is a promising way forward for agents built on frozen models to continually improve over time.

## 2 RELATED WORK

There is a long literature of work on agents that can navigate complex environments. A common approach is to use reinforcement learning (RL), e.g., DRRN (He et al., 2015), KG-A2C (Ammanabrolu & Hausknecht, 2020), CALM (Yao et al., 2020), where agents learn a task over repeated trials. However, while effective, such agents typically require a large number of trials to learn and have trouble adapting to unexpected changes in the test environment. More recently, (Adaptive-Agent-Team et al., 2023) demonstrated AdA, an agent that could rapidly adapt to open-ended novel 3D problems, using meta-reinforcement learning, essentially being able to change its policy on the fly. However, AdA required vast amounts of pretraining, and this skill was still limited to the style of environments and problems seen in pretraining.

Recently, LLMs have provided a new tool for building goal-directed agents (Huang et al., 2022). Given a linguistic description of the world state, a task, and a history, the LLM can be prompted to suggest next actions to take to achieve a goal, exploiting their wealth of semantic knowledge about

the world and requiring little training, e.g., SayCan (Ahn et al., 2022), ReAct (Yao et al., 2022), and more recently SwiftSage (Lin et al., 2023), which combines a supervised agent and a deliberative agent together. However, while performing reasonably with little training data, such agents are unable to learn and adapt from experience.

Two recent systems have demonstrated how a frozen-model-based agent could improve at a task. Voyager (Wang et al., 2023) operates in the world of Minecraft, growing a (code-based) skill library from rich feedback of its failures. Reflexion (Shinn et al., 2023) improves at a task by *reflecting* on a failed attempt at that task and devise a new plan that accounted for that mistake, used in the subsequent prompt to retry the task. While Reflexion did not have a long-term memory, and its reflections were task- and environment-specific, e.g., "In the next trial, I will go to desk 1 and find the lamp.", we take inspiration from it to build an agent, CLIN, which continually maintains and adapts a long-term, persistent memory of reflections, useful across different trials, tasks, and environments.

More generally, others have found that a memory of useful learnings can be used to improve frozen LLM behavior, e.g., in QA (Dalvi et al., 2022; Tandon et al., 2022; Madaan et al., 2023), or for modeling social behavior (Park et al., 2023). We apply this finding to goal-directed agents.

Finally, we note that the *content* of experiential memory is also important. Specifically, CLIN learns a memory of *causal abstractions*, which can be seen as learning a linguistic form of action model, describing the causal effects of actions. While there has been substantial work in the planning community of learning action models in a fully formalized context (Arora et al., 2018; Aineto et al., 2018), CLIN loosely applies this idea in the linguistic world of LLM agents.

## 3 APPROACH

**Problem Formulation.**     Sequential decision-making tasks require agents to repeatedly observe and then act in an environment until they accomplish a specific goal. At a high level, this can be accomplished by developing beliefs about the world, acting on the environment based on those beliefs, and then updating one's beliefs based on the observed outcome. Here, we investigate constructing an agent that can continually update its beliefs through interaction and observation while exploiting its past experience toward solving unseen parametric variations of tasks.

**Setup.**     We investigate our continual learning agents in simulated environments. Our environments are modeled in a high-fidelity text-based simulator (Wang et al., 2022), where both actions and observations are expressed in natural language. Let's define the task space to be $\mathcal{M}$, a collection of partially observable Markov Decision Processes (POMDPs) that can be executed in a set of environment configurations $\mathcal{E}$. Each task $m \in \mathcal{M}$ has an initial state and a desired winning state, which vary depending on the environment $e \in \mathcal{E}$.

Our setup allows an agent to attempt a task several times; each time is denoted as a *trial*, $\mathcal{T}$, which consists of a total of $\tau$ steps. Each step comprises an action by the agent ($a$), and in response, the simulator returns the result of that action in the form of an observation ($o$) and a reward ($r$). A collection of $K$ trials is called an *episode*. The environment gets reset when the task reaches an end state (such as completing, failing, or timing out). In our continual learning setup, the agent retains its memory across trials/episodes, reaping the benefits of continued interaction with the environment.

### 3.1 **CLIN**: A GENERATIVE AGENT CONTINUALLY LEARNING FROM INTERACTIONS

To act in the world, CLIN uses three modules: a **memory**, a **controller**, and an **executor**, illustrated in Figure 2 and which we now describe. Learning then occurs using a fourth module, a **memory generator**, to generate an updated memory after each trial.

**Memory.**     CLIN's memory ($\mathcal{S}$) is a persistent, dynamic collection of NL sentences that express causal abstractions, generated by reflecting on past experiences in order to help the agent perform better in the future. This generation process is described shortly in Section 3.2. For example, having an explicit causal insight, "opening the fridge is `necessary` to access apple juice", learned from past experiences, can reduce the action search space for CLIN while looking for "apple juice" in the same environment in future trials. To aid in continual learning, the memory also captures *mistakes* made by the agent in previous trials, similar to reflective agents explored in recent work (Shinn et al., 2023; Park et al., 2023), noting actions that failed to contribute to a task.

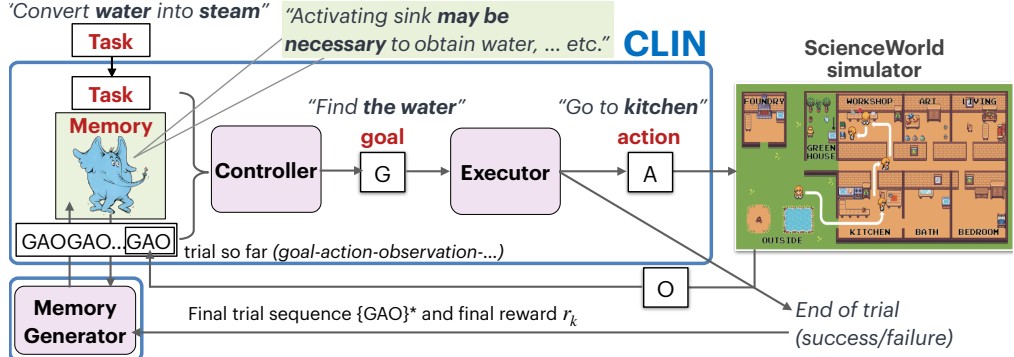

Figure 2: The architecture of CLIN. A **controller** takes the current task, retrievals from memory, and the trial so far, to generate the next goal to achieve. The **executor** then converts this to a valid action to perform towards that goal. The simulator then performs the action and returns an observation of that action's effect. Memory is updated at the end of each trial by the **memory generator** (Section 3.2).

**Controller.** The role of the controller is to generate the next goal to pursue in service of the task. In CLIN it is a frozen LLM, whose prompt includes the current **task** $m$, e.g., "convert water into steam", retrievals from the **current memory** $\mathcal{S}$, and the **trial so far** (the sequence of goal-action-observation triples $\{g_1, a_1, o_1, ..., g_t, a_t, o_t\}$), and is prompted to output the next **goal** $g_{t+1}$ to pursue, e.g., "find water". Memory items are retrieved using both the task instruction and the trial history. The controller first selects one or more memory items given the current state and if they are useful for the next action to progress in the task. After that, it appends the learning, if selected, in context to generate a goal, otherwise the goal is generated based on the trial history (see full prompt in Figure 6).

**Executor.** The role of the executor is to convert the generated goal $g_{t+1}$ into a valid **action** $a_{t+1}$ that can be executed in the environment in pursuit of that goal. Again a (frozen) LLM is used, whose prompt includes the goal $g_{t+1}$, the trial so far, and all the possible actions that can be performed in the current state (provided by the simulator, as is standard practice in current generative agent research (Ahn et al., 2022; Yao et al., 2022; Lin et al., 2023; Park et al., 2023)). The list of possible actions is expressed as possible action templates and available objects that can instantiate them, rather than a combinatorially large enumeration of possible actions. The model is then prompted to generate a candidate action to perform (see prompt in Figure 6). Finally, CLIN checks this candidate action is one of the valid actions. If it is not, it finds the most similar valid action using the pre-trained embeddings from the `sentence-transformer` model (Reimers & Gurevych, 2019). If the top-ranked valid action has a similarity score greater than a threshold (here, 0.9, chosen as a hyperparameter), the action is selected. Otherwise, we perform iterative refinement (Madaan et al., 2023) by suffixing the context with feedback that the generated candidate action is not executable. This allows the executor to retry the generation for up to a maximum number of tries (here, 5).

Finally, upon executing the action $a_{t+1}$, CLIN receives a partial next state, as an **observation**, from the simulator and the reward $(r) \in [0, 1]$. Rewards are nominally given by the simulator for achieving either major task subgoals (e.g., finding water, for the boiling task), or minor and optional subgoals (e.g., being in the kitchen, for the boiling task). Rewards are sparse and generally only supplied after the completion of a task subgoal. A snapshot of a full trial is given in lines 4-10 in Algorithm 1.

Note that CLIN does not make use of any gold data to identify goals and memories. Rather, we expect CLIN to perform a balanced act of exploration-exploitation by interacting, learning, and adapting to unseen tasks or environment configurations—a key difference from few-shot generative agents by previous work (Ahn et al., 2022; Yao et al., 2022; Lin et al., 2023; Park et al., 2023).

### 3.2 MEMORY: A COLLECTION OF ADAPTIVE CAUSAL ABSTRACTIONS

At the end of each trial (completion or failure), CLIN uses a **memory generator** to create or update its memory. The memory generator is a (frozen) LLM prompted to reflect on the current trial and memory, and generate a new memory of insights in the form of (English sentences expressing) useful **causal abstractions**, as we now describe.

**Algorithm 1** Continual Learning with CLIN

1: **procedure** ADAPTATION(Task: $m$, Env: $e$, Memory: $\mathcal{S}$):
2:     Initialize Memory: $\mathcal{S}_0$
3:     **for** $k \in 1, \cdots, K$ **do**:
4:         Intialize Trial $\mathcal{T}$, $t$
5:         **while** $t <$ max. steps or task not complete **do**:
6:             $g_t = $ Controller $(m, e, \mathcal{T}_{<t}, \mathcal{S}_{k-1})$
7:             $a_t = $ Executor $(g_t, $ admissible actions$)$
8:             $r_t, o_t = $ Simulator $(\mathcal{T}_{<t}, a_t)$
9:             $\mathcal{T}_{<t+1} = \mathcal{T}_{<t} + (g_t, a_t, o_t, r_t)$
10:        Final reward $r_k = r_t$
11:        $\mathcal{S}_k = $ memory-generator $(\{\mathcal{S}_{<k}\}, \mathcal{T}_k, r_k)$
12:
13: **procedure** GENERALIZATION(Task: $m$, Env: $e$, past $m'/e'$)
14:     $\{\mathcal{S}_{\text{crucial}}, r_k\} = $ crucial-memories (past $m'/e'$)
15:     $\mathcal{S}_{\text{meta}} = $ meta-memory $(\{\mathcal{S}_{\text{crucial}}, r_k\}, m)$
16:     ADAPTATION$(m, e, \mathcal{S}_{\text{meta}})$

| | X (actions) | *relation* | Y (actions) |
|---|---|---|---|
| **Memory** $\mathcal{S}_1$ end of trial 1 | Using the lighter on the metal pot | **may be necessary** | to heat the water in the pot |

**ADAPTATION**

| | X (actions) | *relation* | Y (actions) |
|---|---|---|---|
| **Memory** $\mathcal{S}_5$ end of trial 5 | Using the lighter on the metal pot | **should be necessary** | to heat the water in the pot |
| | Turning on the stove | **does not contribute** | to boiling if the stove is broken |

| | X (actions) | *relation* | Y (actions) |
|---|---|---|---|
| **Memory** $\mathcal{S}_5$ best in Env 1 | Using the lighter on the metal pot | **should be necessary** | to heat the water in the pot |
| **Memory** $\mathcal{S}_3$ best in Env 2 | Turning on the stove | **should be necessary** | to create a heat source |

**GENERALIZATION**

| | X (actions) | *relation* | Y (actions) |
|---|---|---|---|
| **Meta memory** for Env 3 | Using a heat source (stove, lighter) on the container | **should be necessary** | to heat a substance |

Figure 3: (LEFT) CLIN's continual learning algorithm. (RIGHT) Example causal abstractions.

Learning about state transitions is essential for sequential decision-making tasks (Mnih et al., 2013), which can be manifested by knowing 1) actions enabling desired state transitions, 2) actions producing undesired or no change in states, and 3) state transitions that contribute to the task progress. To generate these kinds of knowledge, the generator is prompted to generate insights in a particular syntax (see prompt in Figure 7). To capture good actions enabling desired changes and helpful state transitions, we use the template "X is NECESSARY to Y", and to capture contrastive examples of unsuitable actions and state transitions, we employ "X DOES NOT CONTRIBUTE to Y", as depicted in Section 4.3, where X, Y are related to actions. These abstractions are functionally analogous to hindsight experience replay (Andrychowicz et al., 2017), obtained from CLIN's past self-explorations.

While useful insights can be abstracted from the trials, CLIN's exploration can be limited, especially in the early trials, given an incredibly large action space. Hence, incomplete exploration can pose varying degrees of uncertainty on extracted insights. To capture this, we also include a measure of uncertainty in each abstraction by either of the two linguistic variations in their surface forms: "X may ..." to denote moderate to high uncertainty, and "X should ..." to indicate low uncertainty (See Section 4.3). In the course of continual learning, as CLIN gathers experience, we expect the level of uncertainty to change depending on the frequency of their use and their fitness to the task.

**Updating Memory Across Trials.** CLIN continually attempts to solve a task in an environment for multiple trials (in sum, an episode). To update the memory after each trial within an episode, the memory generator is prompted with the most recent trial (a sequence of $(g_t, a_t, o_t)$ tuples and the final reward $r_k$[1]), and the memories from the three most recent trials $\{\mathcal{S}_{k-2}, \mathcal{S}_{k-1}, \mathcal{S}_k\}$. It is then prompted to generate an updated memory $\mathcal{S}_{k+1}$, namely a new list of semi-structured causal abstractions in the forms described above, for use in the next trial. Although we do not specify a maximum size for the memory, we observe that size of the generated memory (i.e., the number of causal abstractions generated) is far less than the number of actions executed in the trial, indicating the memory-generator additionally performs a saliency-based pruning to keep only important insights based on the success of the trial, as indicated by the final reward $r_k$ at the end of the trial $\mathcal{T}_k$.

### 3.3 META-MEMORY FOR BETTER GENERALIZATION

Updating memory based on past insights and the current trial to influence future trials for the same task in the current environment configuration during test-time adaptation. However, to generalize across tasks or environment configurations, the memory needs to contain more generalized causal

---

[1]The reward is converted to NL feedback for a LLM using 7 simple rules, e.g., "if score >= 0 and score < 20 then feedback = *"The agent performed poorly and made some progress but not enough to solve the task."*

abstractions than memories used across trials in an episode. We call this as **meta-memory**, abstracted across multiple episodes of solving different tasks in different environment configurations to be applicable in future episodes.

**Auto-curriculum Selection.** Before we generate the meta-memory, it is important to choose memories extracted from the best trials from previous episodes because random sampling may not benefit CLIN in zero-shot generalization (Adaptive-Agent-Team et al., 2023). Following the prioritized level replay scheme (Jiang et al., 2021), we choose the most successful trial per episode and retrieve memories abstracted from those trials with a fixed archive of size 10, a hyperparameter.

**Generating Meta-Memory.** The goal of the meta-memory is to help CLIN generalize to unseen tasks and/or environments. While we keep the format of the causal abstractions the same as memories generated across trials, the prompt to generate the meta-memory is different than those used for generating per-trial memory. When the new memory is to be used for the *same* task but in a *different* environment, the prompt instructs for a meta-memory helpful "to solve the same task in a new environment configuration" given the target task description with an expectation that meta-memory abstractions will entail generic causal insights about the task irrespective of environment configurations (see Figure 8). Similarly, when the new memory is to be used for a *different* task, the prompt is modified accordingly to reflect this (Figure 9). Along with the target task description for better memory generation, each past memory selected to generate the meta-memory is attached to the final rewards for the associated trials, allowing the generator to combine insights across episodes and assign the levels of uncertainty using the evidence of success.

## 4 RESULTS AND ANALYSIS

**Experimental Setup.** Test-time adaptation and generalization via continual learning require a variety of complex tasks and environment configurations to allow an agent to explore, learn latent causal insights from interactions, and exploit them in the future. We choose ScienceWorld (Wang et al., 2022), a text-based interactive environment requiring complex interactive reasoning processes to solve a plethora of science-theory-based tasks spanning several diverse classes (e.g., thermodynamics, genetics, friction, etc.). The virtual space presents 10 sub-places: foundry, greenhouse, outside area, an art studio, workshop, kitchen, living room, bedroom, bathroom, and a hallway connecting inside rooms. The presence of several objects, their individual states, and action templates renders the search space intractable for any agent. ScienceWorld presents strikingly different environment configurations across task types, making it a rich testbed for evaluating adaptation and generalization. ScienceWorld tasks are partitioned into Short (S), e.g., *pick & place* and Long (L), e.g., *grow plant*, tasks based on the number of required actions to succeed.

Here, we define our setups for zero-shot adaptation (ADAPT) and generalization (GEN-ENV and GEN-TASK). For all setups, we test CLIN and competing baselines on 18 tasks (two task instances from 9 classes) in several environment configurations from the test split of the ScienceWorld benchmark resulting in a total of 164 task-environment combinations unless stated otherwise. We evaluate based on the final reward score provided by the ScienceWorld simulator.

**ADAPT**: This setup focuses on CLIN's ability to adapt to a task by attempting it for several trials in the same environment configuration. Most importantly, CLIN initializes with an empty memory at the beginning of the first trial and generates memory at the end of each trial. While the environment gets reset at the trial boundary, CLIN's memory continues to be updated, capturing informative causal abstractions pertaining to both successful and failed actions. Here, we compare with Reflexion (Shinn et al., 2023), a SOTA, however, CLIN differs from Reflexion by how the memory is abstracted.

**GEN-ENV**: In this setup, we focus on CLIN's ability to transfer its learning from past experiences to solve tasks in an unseen environment. For a task $m$, we run CLIN for 10 different (train) environment settings (with varying objects and starting locations) and then create meta-memories from its exploration to solve the same task in an unseen (test) environment. Here, we compare CLIN with RL methods DRRN (He et al., 2015), KG-A2C (Ammanabrolu & Hausknecht, 2020), and CALM (Yao et al., 2020) trained on all (large) training variations with simulator reward and Generative Language agents, SayCan (Ahn et al., 2022), ReAct (Yao et al., 2022), and Reflexion (Shinn et al., 2023), prompted with few-shot demonstrations.

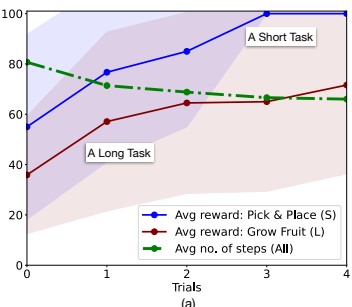
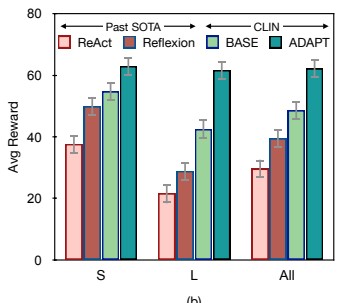

| | | | |
|---|---|---|---|
| **CLIN's ADAPT improvements** | | | |
| **Type** | **#trials to success (↓)** | | **%ep. improv.** |
| S | 3.3 | | 29.2 |
| L | 3.2 | | 37.2 |
| All | 3.3 | | 33.2 |

(c)

Figure 4: **Rapid task adaptation with CLIN. (a)** Example tasks where CLIN improves scores across trials. For CLIN, Trial-0 is the BASE, Trial-4 is the ADAPT. **(b)** Comparison of CLIN with Reflexion (Shinn et al., 2023). **(c)** CLIN improves from BASE to ADAPT (full results in Appendix C).

**GEN-TASK**: In this setup, we focus on CLIN's ability to transfer its learning from past experiences to solve a new task in the same environment. For an environment $e$, we run CLIN for to solve a task $m$ and then condense its learning to solve a novel task $m'$ in the environment $e$. We took all test examples where we have a different task defined in the same environment configuration. (Adaptive-Agent-Team et al., 2023) suggests that transferring learning from a random task can be very hard; hence we couple tasks that are related (revolve around overlapping task-critical objects/locations such water, kitchen), such as *boil* and *freeze* to measure transfer learning from one to the other. This is a novel setup where we do not have any off-the-shelf baselines. However, here, we compare against CLIN-BASE, a strong baseline agent.

**GEN-ADAPT (G+A)**: If CLIN, in GEN-ENV or GEN-TASK setting, does not successfully complete the new task, it can continue learning and retrying that task. We refer to this setup as GEN-ADAPT. CLIN can use any instruction-tuned LLM (Chung et al., 2022) as part of the controller, executor, and memory generator. In this paper, we use `gpt-4`, the same as our generative agent baselines.

### 4.1 CLIN EXHIBITS RAPID TASK ADAPTATION

Figure 4a demonstrates two example trends where CLIN learns from its own prior attempts (ADAPT) and gets better at solving a given task. Apart from length, the difficulty level of a task also depends on the environment configuration (hence, variance across environment configurations for each task). CLIN quickly adapts to a short task, *Pick & Place*, solving it in its 4th attempt, whereas for a longer task, *Grow Fruit*, it is not solved after 5th (max) attempts. Furthermore, Figure 4a depicts, CLIN becomes more efficient in later trials by solving the tasks with a lower number of (average) steps. Figure 4c shows an average number of attempts[2] for CLIN to solve a task and % episodes per task where scores improved compared to its own first trial.

Next, we compare CLIN with Reflexion, the reflective SOTA agent, in Figure 4b. CLIN already starts off with a stronger base performance (see discussion in 4.3), however, CLIN's relative improvement in ADAPT is significantly stronger than Reflexion's gain from its base agent ReAct. Furthermore, CLIN's relative improvement is higher for longer tasks. This can be attributed to CLIN's persistent memory, which gets refined over past trials, whereas Reflexion may fall short of collecting useful learnings from earlier trials as it only focuses on the current trial for its reflections (hence not long-term). Furthermore, CLIN accumulates both useful (for the task) and harmful (for the task) causal learnings, whereas Reflexion only learns from its mistakes, lacking comprehensive learning.

### 4.2 CLIN OUTPERFORMS SOTA, GENERALIZING TO NOVEL ENVIRONMENTS AND TASKS

**Generalizing to new environments.** Table 1 compares CLIN with baselines that learn from training environmental variants for a task to improve its performance in a novel environment [3]. Language agents (including CLIN) that use NL feedback from the ScienceWorld (e.g., "Door to the kitchen is closed") perform significantly better compared to RL methods that purely rely on

---

[2]During ADAPT, CLIN tries up to 5 trials. If it solves a task with a score of 100, it stops retrying.

[3]Baseline numbers are derived from Table 1 in (Lin et al., 2023)

| Task | Type | RL Methods | | | Generative Language Agents | | | CLIN (ours) | | |
|------|------|------|-------|------|--------|-------|-----------|------|---------|------|
| | | DRRN | KGA2C | CALM | SayCan | ReAct | Reflexion | BASE | GEN-ENV | G+A |
| Temp | S | 6.6 | 6.0 | 1.0 | **26.4** | 7.2 | 5.9 | 25.2 | 15.7 | 13.8 |
| Temp | S | 5.5 | 11.0 | 1.0 | 8.0 | 6.1 | 28.6 | 53.2 | 49.7 | **58.2** |
| Pick&Place | S | 15.0 | 18.0 | 10.0 | 22.9 | 26.7 | 64.9 | 92.5 | 59.2 | **100.0** |
| Pick&Place | S | 21.7 | 16.0 | 10.0 | 20.9 | 53.3 | 16.4 | 55.0 | **100.0** | **100.0** |
| Chemistry | S | 15.8 | 17.0 | 3.0 | 47.8 | 51.0 | **70.4** | 44.5 | 42.2 | 51.7 |
| Chemistry | S | 26.7 | 19.0 | 6.0 | 39.3 | 58.9 | 70.7 | 56.7 | 85.6 | **93.3** |
| Lifespan | S | 50.0 | 43.0 | 6.0 | 80.0 | 60.0 | **100.0** | 85.0 | 65.0 | **100.0** |
| Lifespan | S | 50.0 | 32.0 | 10.0 | 67.5 | 67.5 | 84.4 | 70.0 | 75.0 | **90.0** |
| Biology | S | 8.0 | 10.0 | 0.0 | 16.0 | 8.0 | 8.0 | 10.0 | 32.0 | **32.0** |
| Boil | L | 3.5 | 0.0 | 0.0 | **33.1** | 3.5 | 4.2 | 7.0 | 4.4 | 16.3 |
| Freeze | L | 0.0 | 4.0 | 0.0 | 3.9 | 7.8 | 7.8 | **10.0** | 8.9 | **10.0** |
| GrowPlant | L | 8.0 | 6.0 | 2.0 | 9.9 | 9.1 | 7.3 | 10.2 | 10.9 | **11.2** |
| GrowFruit | L | 14.3 | 11.0 | 4.0 | 13.9 | 18.6 | 13.0 | 35.9 | 70.8 | **94.5** |
| Biology | L | 21.0 | 5.0 | 4.0 | 20.9 | 27.7 | 2.6 | 70.0 | 42.8 | **85.6** |
| Force | L | 10.0 | 4.0 | 0.0 | 21.9 | 40.5 | 50.6 | 53.5 | 70.0 | **100.0** |
| Friction | L | 10.0 | 4.0 | 3.0 | 32.3 | 44.0 | **100.0** | 56.5 | 70.0 | 94.0 |
| Genetics | L | 16.8 | 11.0 | 2.0 | 67.5 | 25.7 | 50.9 | 77.4 | 84.5 | **100.0** |
| Genetics | L | 17.0 | 11.0 | 2.0 | 59.5 | 16.8 | 23.7 | 62.3 | 61.4 | **100.0** |
| | S | 22.1 | 19.1 | 5.2 | 36.5 | 37.6 | 49.9 | 54.7 | 58.3 | **71.0** |
| | L | 11.2 | 6.2 | 1.9 | 29.2 | 21.5 | 28.9 | 42.5 | 47.1 | **68.0** |
| | All | 16.7 | 12.7 | 3.6 | 32.9 | 29.6 | 39.4 | 48.6 | 52.7 | **69.5** |

Table 1: Comparing CLIN with baselines for **generalization across unseen environments**

(sparse) numeric rewards from the environment to learn a policy. We observe a positive generalization effect in GEN-ENV (average 4 point gain) compared to BASE where CLIN tries to solve the tasks zero-shot. With a strong BASE performance, CLIN beats all baselines in generalization performance. Furthermore, in G+A, CLIN shows a substantial 16 additional improvement, beating the SOTA reflective agent by 23 points. Figure 5a additionally shows trend of improvement compared to when CLIN does not start with a meta-memory. Meta-memory helps CLIN with a stronger start than BASE (52.7 vs. 48.6), with a continued gain in scores till the end of Trial-4 (G+A: 69.5 vs. ADAPT: 62.2). The stronger start for CLIN with meta-memory also results in fewer steps to solve a task. Unlike imitation learning-based agents, TDT (?) and SwiftSage (Lin et al., 2023), CLIN (and most baselines) does not use any gold trajectories. Refining its memory only from self-generated trajectories, CLIN outperforms TDT on all 18 tasks and SwiftSage on 8/18 (mostly long) tasks.

**Generalizing to new tasks.** Mirroring trends from GEN-ENV, CLIN demonstrates strong transfer learning to new tasks with 13-point improvement over its BASE performance, being better at 38.8% of times (Figure 5c). The improvement attributes to critical learning about the environment ("apple juice is in the fridge", required for both boiling and freezing it), leading to improvement in previously low-performing tasks in both ADAPT and GEN-ENV setups. This transfer learning in GEN-TASK and G+A helps CLIN to solve the tasks with a lesser number of steps[4] and achieve higher rewards.

### 4.3 DISCUSSION

**Importance of memory structure.** CLIN extracts causal abstractions structured around 'necessary' and 'does not contribute' relations. To ablate, we modified our memory generator to generate free-form advice for future trials, which, however, ended up generating generic insights without any causal abstractions (Figure 13). The average reward drops by 6 points (in 10% cases than CLIN) when using the unstructured memory, indicating the usefulness of causal abstractions, as shown in Figure 5d.

**Superior BASE performance.** Figure 4 depicts a superior BASE performance for CLIN than the final performance of both ReAct and Reflexion despite using the same underlying LLM (here, `gpt-4`). We find if we ablate for the controller module in CLIN, responsible for generating a goal before outputting the next action, CLIN's BASE performance drops in 44% cases. With an 18 point drop in average reward (see Figure 5d), Abl-Contoller-BASE version of CLIN becomes equivalent to ReAct, the base agent for Reflexion, demonstrating the importance of controller even in BASE setup.

**A qualitative example.** Figure 3 depicts how memory items get refined during task adaptation and for generalization for a task *boil*. Env2 has a working stove, whereas in Env1, the stove is broken, but

---

[4]# steps in Figure 5(a),(b) are normalized between 0-1, 1 being maximum #steps allowed for a task.

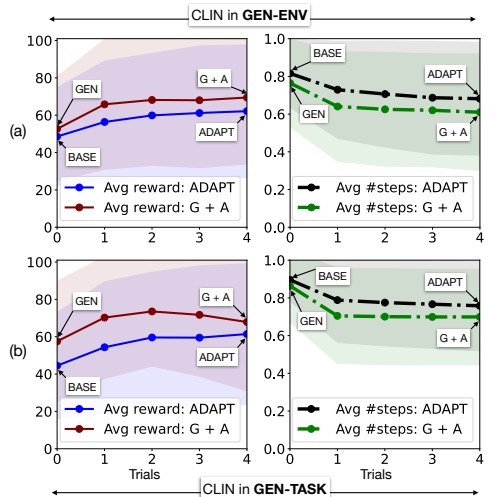

| | **CLIN's GEN-TASK improvements** | | | |
|---|---|---|---|---|
| **Type** | **GEN-TASK** | | **G + A** | |
| | $\triangle$**avg score** | **%ep. improv.** | $\triangle$**avg score** | **%ep. improv.** |
| S | 14.6 | 40.0 | 4.9 | 5.7 |
| L | 10.3 | 36.7 | 9.2 | 15.6 |
| All | 13.0 | 38.8 | 6.5 | 9.3 |

(c)

| **Ablations for CLIN** | | |
|---|---|---|
| **Ablation Setup** | $\triangle$**avg score ($\downarrow$)** | **%ep. drop. ($\uparrow$)** |
| Abl-Causal-Memory | -6.2 | 10 |
| Abl-Controller-BASE | -18.1 | 44.8 |

(d)

Figure 5: Reward and #steps trends for CLIN in **(a)** GEN-ENV and **(b)** GEN-TASK. **(c)** % episode improvements and score change than CLIN without meta-memory (GEN-TASK). **(d)** CLIN ablations.

a lighter is available as an alternative. With a number of trials in these environments, CLIN learns how to use these two devices to generate heat. In an unseen environment with a broken stove, CLIN quickly receives a positive reward by using a lighter to heat a substance. While insights within an episode are often specific, e.g., "Using the *lighter* on the metal pot should be necessary to heat the *water* in the pot", CLIN compiles these insights for a new target environment (as meta-memory), e.g., "Using a *heat source* (stove, lighter) on the container should be necessary to heat a *substance*." Appendix B contains examples of memories generated during adaptation and generalization.

**Limitation: Lack of exploration.** CLIN's learnings are dependent on its own past experience. If CLIN never explores a location in the environment or does not perform an action, an insight related to the unobserved activity can never be generated. Hence, exploration becomes important when task-critical location or action in unknown to CLIN from past trials. For example, in task of creating an orange paint, the agent is supposed to find red and yellow paints from the art studio. However, art studio is not visible when CLIN starts from location 'outside'. Unless the CLIN knows that there exists an art studio, it tries alternative method to create orange paints from other irrelevant objects (e.g., an orange) and remains unsuccessful. When a memory related art-studio appears from past exploration, CLIN is able to successfully complete the task. Similarly, in boil or freeze tasks, CLIN is unable to perform well which requires it to consistently measure the temperature of the substance to know its boiling/freezing point—an act it could never perform successfully in past trials resulting into less useful memory insights and subsequent lower performance in future trials.

**Limitation: Poor memory retrieval.** For a task of boiling gallium, CLIN is supposed to use oven/blast furnace and not a stove. In the meta-memory for boiling tasks, there are two insights regarding the act of boiling: "Activating stove should be necessary to boil a substance" and "Using an alternative heat source (e.g., oven or fire pit) may be necessary if the initial heat source is insufficient." However, CLIN repeatedly retrieves the former and hence failing at the task despite performing other actions (e.g., finding gallium) correctly. This problem intensifies at the initial trial during generalization due to the presence of insights with varied initial conditions for them to be applied. This can be circumvented by improved memory representation, which we leave as a future work.

## 5 CONCLUSION

Our goal is a system that can continually improve over time, both while rapidly adapting to a task by multiple retries and efficiently generalizing to novel tasks and environments. We propose CLIN, an architecture for language agents that constructs a persistent, dynamic memory of causal abstractions, refines it over time and uses it effectively to improve its performance on future tasks, achieving state-of-the-art performance. Our work systematically evaluates a novel nonparametric learning paradigm, promising never-ending learning abilities to frozen language agents.

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

## A    CLIN PROMPTS

Figures 6 to 9 are the complete prompts for next-action generation (controller + executor), memory-generator during ADAPT, GEN-ENV, and GEN-TASK.

```
[System]: You are an AI agent helping execute a science experiment in a simulated
environment with limited number of objects and actions available at each step.

[User]:
Possible objects ( value an OBJ can take ):
{objects_str}

Your next action should be in one of the following formats:
Possible actions:
{actions_str}

If I say \"Ambiguous request\", your action might mean multiple things. In that case,
respond with the number corresponding to the action you want to take.

What action would you like to do next?

First, scan the (unordered) list of learnings, if provided. Decide if any of the
learnings are applicable given the last observation to make progress in this task. Then
only use selected learnings, if any, to construct a rationale for picking the next
action. If no Learning is selected, construct the rationale based on the last
observation. Format your response as follows:

Write 'I used learning id(s):' as a comma separated list; the list can be empty if no
learnings selected. Then, write $$$ followed by the rationale. Finally, write ###
followed by the single next action you would like to take.

If you think you have completed the task, please write TASK_COMPLETE as the next action.

If the task requires you to 'focus' on something (OBJ), please write FOCUS ON <OBJ> as
the next action. FOCUS is a extremely critical action that can be only used the number
of times 'focus' is mentioned in the task description. Using it more than that or
inappropiately (such as on a wrong object) will terminate the session and the task will
be rendered as incomplete.

If you performed an action that requires waiting to see the effect, please write 'wait'
as the next action.
```

Figure 6: Prompt for the Controller and the Executor

## B    EXAMPLE MEMORIES

Example generated memory for ADAPT, GEN-ENV, and GEN-TASKsetups in Figures 10 to 12.

```
[System]: You are an expert assistant.

[User]:
You are given CURRENT TRACE, a sequence of actions that an agent made in a world to
accomplish a task.

Task is detailed at the beginning.
For each action, there is a rationale why the agent made that action.
There is an observation that provide details about the new state of the world after
each action was executed.
The CURRENT TRACE is accompanied by an EVALUATION REPORT indicating the success of the
attempt to the task.

You can also be provided with PREVIOUS LEARNINGS which are learnings from the previous
attempts by the agent for the same task in the same environment/world. TASK indicates
the task description. EPISODE indicates the number of previous attempts of the task.

Generate a summary of learning, as a numbered list, that will help the agent to
successfully accomplish the SAME task AGAIN, in the SAME world.

Each numbered item in the summary can ONLY be of the form:
X MAY BE NECCESSARY to Y.
X SHOULD BE NECCESSARY to Y.
X MAY BE CONTRIBUTE to Y.
X DOES NOT CONTRIBUTE to Y.

{CURRENT TRACE}
Action: ...
Observation: ...
...
EVALUATION REPORT:
REWARD_FINAL: 100. This means: The agent has performed exceptionally well and
successfully solved the task.

Summary of learning as a numbered list:
```

Figure 7: Prompt for CLIN's memory generator during ADAPT

```
[System]: You are an expert assistant.

[User]: You are given a collection of learning lists, that are derived from actions
made by an agent and subsequent observations from a world to accomplish a TYPE of TASKs.
All of these TASKs belong to a same TYPE (such as 'boiling') but they are executed in
different ENVIRONMENT configurations. A different ENVIRONMENT configuration means there
are presence of a different set of objects (lighter instead of a stove) that are
critical for solving the TASK, presence of a different set of distractor objects that
are not useful for the TASK, a different floor plan, etc.

For each learning list, the TASK description is provided at the beginning as TASK:

Each learning list indicates a list of learnings from the agent's best attempt to solve
the TASK.

Each learning list is associated with an EVALUATION REPORT indicated how sucessful the
respective attempt was for solving the task.

Consider all learning lists and combine them in to a summary of learnings, as a
numbered list, that will help the agent to successfully accomplish a NEW TASK related
to the previous TASKs (such as 'boliing') in an ENVIRONMENT configuration that it has
not seen before. The NEW TASK description will be provided.

Each numbered item in the summary can ONLY be of the form:
X MAY BE NECCESSARY to Y.
X SHOULD BE NECCESSARY to Y.
X MAY NOT CONTRIBUTE to Y.
X DOES NOT CONTRIBUTE to Y.

{PREVIOUS LEARNINGS}
TASK: ...
LEARNINGS:...
EVALUATION REPORT:
REWARD_FINAL: 100. This means: The agent has performed exceptionally well and
successfully solved the task.
...

NEW TASK: ...
Summary of learning as a numbered list:
```

Figure 8: Prompt for CLIN's memory generator during GEN-ENV

```
[System]: You are an expert assistant.

[User]: You may be given a list of learnings, that are derived from actions made by an
agent and subsequent observations from a world to accomplish a TASK in an ENVIRONMENT
CONFIGURATION.

For the learning list, the TASK description is provided at the beginning as TASK:

The learnings are from the agent's best attempt to solve the TASK.

The learning list is associated with an EVALUATION REPORT indicated how sucessful the
attempt was for solving the task.

Now, generate a summary of learnings from the existing ones if provided, such that they
will be useful to the NEW TASK in the SAME ENVIRONMENT CONFIGURATION. The NEW TASK may
require different actions which are not captured in the given learnings but given
learnings can be used to infer about the ENVIRONMENT CONFIGURATION. The NEW TASK
description will be given. If PREVIOUS LEARNINGS says 'No learnings available',
improvise learnings for the NEW TASK.

Each numbered item in the summary can ONLY be of the form:
X MAY BE NECCESSARY to Y.
X SHOULD BE NECCESSARY to Y.
X MAY NOT CONTRIBUTE to Y.
X DOES NOT CONTRIBUTE to Y.

{PREVIOUS LEARNINGS}
TASK: ...
LEARNINGS:...
EVALUATION REPORT:
REWARD_FINAL: 100. This means: The agent has performed exceptionally well and
successfully solved the task.
...

NEW TASK: ...
Summary of learning as a numbered list:
```

Figure 9: Prompt for CLIN's memory generator during GEN-TASK

Task: Your task is to determine whether round seed shape is a dominant or recessive trait in the pea plant. If the trait is dominant, focus on the blue box. If the trait is recessive, focus on the orange box.

Memory (ADAPT), after Trial5:
1. Looking around MAY BE NECESSARY to gather information about the environment.
2. Moving to the greenhouse SHOULD BE NECESSARY to find the pea seeds.
3. Picking the jar SHOULD BE NECESSARY to access the pea seeds.
4. Moving the round green pea seed to flower pot 2 SHOULD BE NECESSARY to grow the pea plants.
5. Moving the wrinkly green pea seed to flower pot 3 SHOULD BE NECESSARY to grow the pea plants.
6. Waiting SHOULD BE NECESSARY for the pea plants to grow and show their traits.
7. Looking at the adult pea plant in flower pots 2 and 3 MAY BE NECESSARY to observe the traits of the offspring.
8. Focusing on the blue box SHOULD BE NECESSARY to indicate the trait of the seed shape.
9. Continuously looking at the same environment DOES NOT CONTRIBUTE to the task of determining the dominant or recessive trait in the pea plant.
10. Moving to the workshop DOES NOT CONTRIBUTE to determining the dominant or recessive trait in the pea plant.
11. Picking up the battery or the wires in the workshop DOES NOT CONTRIBUTE to finding the pea seeds.
12. Waiting for longer periods MAY BE NECESSARY to allow the pea plants to fully grow and show their traits.

Figure 10: Memory generated after Trial5 in ADAPT for a Genetics task.

```
Task: Your task is to determine which of the two inclined planes (aluminum, platinum)
has the most friction. After completing your experiment, focus on the inclined plane
with the most friction.

Meta-memory (GEN-ENV):
1. Moving to the hallway SHOULD BE NECESSARY to reach the workshop.
2. Moving to the workshop SHOULD BE NECESSARY to find the block.
3. Picking up the block SHOULD BE NECESSARY to move it to the inclined planes.
4. Placing the block on the first inclined plane (either aluminum or platinum) SHOULD
BE NECESSARY to measure the friction.
5. Activating the stopwatch SHOULD BE NECESSARY to time the experiment.
6. Waiting for a certain period MAY CONTRIBUTE to observing the friction effect.
7. Deactivating the stopwatch SHOULD BE NECESSARY to stop timing the experiment.
8. Moving the block to the second inclined plane (either aluminum or platinum) SHOULD
BE NECESSARY to compare the friction.
9. Activating the stopwatch again SHOULD BE NECESSARY to time the second part of the
experiment.
10. Waiting for a certain period again MAY BE NECESSARY to observe the friction effect.
11. Deactivating the stopwatch again SHOULD BE NECESSARY to stop timing the experiment.
12. Focusing on the inclined plane with the most friction SHOULD BE NECESSARY to
conclude the experiment.
13. Repeating the experiment multiple times MAY BE NECESSARY for more accurate results.
14. Looking around in the initial room multiple times DOES NOT CONTRIBUTE to the task.
15. Moving the block back and forth between the two inclined planes DOES NOT CONTRIBUTE
to the task.
```

Figure 11: Meta-memory used in GEN-ENV for a Friction task.

```
Task: Your task is to freeze mercury. First, focus on the substance. Then, take actions
that will cause it to change its state of matter.

Meta-memory (GEN-TASK):
1. Looking around MAY BE NECESSARY to identify the available resources and the layout
of the environment.
2. Moving to different rooms SHOULD BE NECESSARY to find the tools and materials needed
to change the state of the substance.
3. Picking up items like glass cups or metal pots SHOULD BE NECESSARY to contain the
substance for changing its state.
4. Focusing on the substance SHOULD BE NECESSARY to understand its properties and how
to interact with it.
5. Picking up the thermometer SHOULD BE NECESSARY to monitor the temperature of the
substance.
6. Using the thermometer on the substance SHOULD BE NECESSARY to monitor the progress
of the task.
7. Puring the substance into the container SHOULD BE NECESSARY to prepare it for
cooling.
8. Moving the container to a cooling device SHOULD BE NECESSARY to cool the substance.
9. Waiting for a period of time after cooling the substance SHOULD BE NECESSARY to
allow the substance to change state.
10. Repeatedly checking the temperature of the substance SHOULD BE NECESSARY to monitor
the progress of the task.
11. Activating the stove DOES NOT CONTRIBUTE to the task as it does not progress the
task.
12. Picking up unrelated items like a lighter DOES NOT CONTRIBUTE to the task as it
does not progress the task.
13. Moving to unrelated rooms like the workshop DOES NOT CONTRIBUTE to the task as it
does not progress the task.
14. Teleporting to the kitchen MAY BE NECESSARY for the task as it speeds up the
process of moving between rooms.
15. Using the thermometer multiple times on the substance after it reaches freezing
point DOES NOT CONTRIBUTE to the task as it does not progress the task.
```

Figure 12: Meta-memory used in GEN-TASK for a Freeze task.

```
Task: Your task is to make a red paper.

Ablated memory (GEN-TASK):
1. Avoid repeating actions: The agent performed the "look around" action twice at the
beginning of the task. This is unnecessary and wastes time. Once you have a clear
understanding of your surroundings, move on to the next step.
2. Be mindful of the order of actions: The agent moved to the workshop and read the
instructions after looking around twice. It's better to first read the instructions to
understand the task requirements.
3. Use the correct containers for mixing: The agent used a cup for mixing the
ingredients, which was successful. Continue using suitable containers like a cup or a
bowl for mixing.
4. Be clear with your actions: The agent made an ambiguous request when trying to mix
the cup containing red paint. Be specific with your actions to avoid confusion.
5. Avoid unnecessary movements: The agent teleported to the art studio after picking up
the paper. Plan your actions carefully to avoid unnecessary movements.
6. Use the correct method to combine ingredients: The agent successfully mixed the
ingredients in the cup. Continue using suitable methods to combine the ingredients,
such as using a blender or a food processor.
7. Keep track of your actions: The agent moved the paper to the cup and then back to the
inventory. This is unnecessary and wastes time. Keep track of your actions to avoid
repeating them.
8. Always refer back to the instructions: The agent seemed to forget the instructions
to make the red paper. Always refer back to the instructions to ensure you are
following the correct steps.
```

Figure 13: Meta-memory generated for ablation in GEN-ENV for a Chemistry task.

## C   MORE RESULTS

Full results for CLIN outperforming Reflexion is in Table 3. For ScienceWorld benchmark, we exclude electricity tasks since they deviate from standard electrical conventions, prohibiting us from fairly using LLM agents. We choose the first 10 test variants for each 18 tasks selected. The full list of 18 tasks from the benchmark, with the number of test variants used in parentheses:

grow-plant (10), identify-life-stages-1 (5), grow-fruit (10), measure-melting-point-known-substance (10), mendelian-genetics-unknown-plant (10), chemistry-mix-paint-secondary-color (9), freeze (9), lifespan-longest-lived (10), inclined-plane-determine-angle (10), boil (9), use-thermometer (10), chemistry-mix (8), lifespan-shortest-lived (10), find-plant (10), find-living-thing (10), identify-life-stages-2 (4), mendelian-genetics-known-plant (10), inclined-plane-friction-named-surfaces (10).

Short tasks have oracle lengths less than 37 steps (median), and Long tasks have oracle lengths more than equal to 37 steps.

The map to the short names used for tasks in the paper:

Temp: use-thermometer, measure-melting-point-known-substance; Pick&Place: find-plant, find-living-thing; Chemistry: chemistry-mix, chemistry-mix-paint-secondary-color; Lifespan: lifespan-longest-lived, lifespan-shortest-lived; Biology: identify-life-stages-1, identify-life-stages-2, Boil; Freeze; Grow Plant, Grow Fruit; Force: inclined-plane-determine-angle; Friction: inclined-plane-friction-named-surfaces; Genetics: mendelian-genetics-known-plant, mendelian-genetics-unknown-plant.

**Memory correctness**    While the final performance with memory is indicative of their effectiveness, we performed additional human evaluation of generated memory insights for correctness. For both the Gen-Env and Gen-Task setups, we randomly 10 task-environment combinations to evaluate the correctness of memories used in them, notably the meta-memory used for trial 0 (GEN) and memory adapted for the best trial (GEN-ADAPT). Two annotators rated all the insights (cohen's kappa 0.72) for correctness with reference to the respective gold trajectories. Table 2 shows that it is possible to have some meta-memories that are not applicable for a new task or environments at Trial 0; however, with adaptation, in later trials, the correctness of the memory insights significantly improves, leading to a direct increase in task performance.

| | GEN-ENV (Trial 0) | GEN-ADAPT (Best Trial) | | GEN-TASK (Trial 0) | GEN-ADAPT (Best Trial) |
|---|---|---|---|---|---|
| No. of insights | 100 | 105 | No. of insights | 98 | 107 |
| Correct insights | 72.0% | **91.4%** | Correct insights | 73.9% | **91.1%** |
| Final score | 39.1 | **55.9** | Final score | 43.7 | **58.1** |

Table 2: Human evaluation with two annotators (Cohen Kappa 0.72) for correctness of the memory insights for 10 sampled tasks for both GEN-ENV and GEN-TASK settings.

| Task | Type | Generative L. Agents | | CLIN (ours) | |
| | | ReAct | Reflexion | BASE | ADAPT |
| --- | --- | --- | --- | --- | --- |
| Temp | S | 7.2 | 5.9 | 25.2 | 14.3 |
| Temp | S | 6.1 | 28.6 | 53.2 | 51.8 |
| Pick&Place | S | 26.7 | 64.9 | 92.5 | 100.0 |
| Pick&Place | S | 53.3 | 16.4 | 55.0 | 100.0 |
| Chemistry | S | 51.0 | 70.4 | 44.5 | 44.4 |
| Chemistry | S | 58.9 | 70.7 | 56.7 | 56.7 |
| Lifespan | S | 60.0 | 100.0 | 85.0 | 100.0 |
| Lifespan | S | 67.5 | 84.4 | 70.0 | 90.0 |
| Biology | S | 8.0 | 8.0 | 10.0 | 8.0 |
| Boil | L | 3.5 | 4.2 | 7.0 | 15.2 |
| Freeze | L | 7.8 | 7.8 | 10.0 | 10.0 |
| GrowPlant | L | 9.1 | 7.3 | 10.2 | 11.1 |
| GrowFruit | L | 18.6 | 13.0 | 35.9 | 71.6 |
| Biology | L | 27.7 | 2.6 | 70.0 | 81.0 |
| Force | L | 40.5 | 50.6 | 53.5 | 100.0 |
| Friction | L | 44.0 | 100.0 | 56.5 | 72.5 |
| Genetics | L | 25.7 | 50.9 | 77.4 | 100.0 |
| Genetics | L | 16.8 | 23.7 | 62.3 | 92.6 |
| | S | 37.6 | 49.9 | 54.7 | **62.8** |
| | L | 21.5 | 28.9 | 42.5 | **61.6** |
| | All | 29.6 | 39.4 | 48.6 | **62.2** |

Table 3: Comparing CLIN with baselines for **adaptation**

