# OpenReview forum: "CLIN: A Continually Learning Language Agent for Rapid Task Adaptation and Generalization"
_ICLR.cc/2024/Conference — Submitted to ICLR 2024_

### Official Review · Reviewer_7v8j · 2023-10-31

**Soundness:** 2 fair
**Presentation:** 3 good
**Contribution:** 2 fair
**Rating:** 5
**Confidence:** 4

**Summary:**

This paper proposes a memory architecture, CLIN, that generates causal abstractions as textual memory. A memory generator takes the execution results to update the memory. This enables language-based agents to improve their capability over iterations. The inclusion of meta-memory helps generalization to unseen tasks and environments. The experiments in the ScienceWorld benchmark show that CLIN outperforms the state-of-the-art reflective agent, Reflexion.

**Strengths:**

- CLIN shows that the language agent can continually update and improve its task performance even if they do not update weights.
- This paper shows that it is possible to use prompting to Identify the causal abstractions necessary for agents to change their behaviors or fix previous errors.
- The experiment shows that memory design helps generalization and can efficiently solve some tasks with a few trials.

**Weaknesses:**

- The key component of CLIN is the memory abstraction. However, it is hard to understand the criteria of how to determine the necessary causal abstraction and how to assess the uncertainty of the relevant concepts. While the related prompts are provided in the appendix, it will be hard to extend them without understanding the design criteria or rationale for the memory generator. Also, the accuracy of the generated memory is not evaluated, it is unclear whether the performance improves because the LLM generates the correct memory or just provides more details to the task which may not be causal.
- A causal abstraction doesn’t necessarily need to be structured. The drop in ablation when using the free-form advice suggests the importance of the structured sentence format rather than the causal abstraction. This ablation still doesn’t show the usefulness of causal abstraction.

**Questions:**

The experiment results show that the longer tasks such as growing fruit in Fig 4 are harder for CLIN to adapt. But Fig 4c shows memory improves more episodes for the longer tasks, what are the improvements for long vs. short tasks? What kind of information is still missing for the long tasks?

---

> ### Author Response · Authors · 2023-11-18
> **Response to Reviewer 7v8j**
>
> Thank you for taking the time to review our paper! We were happy to read that you appreciated our main points: that our framework non-parametrically updates its learning through a persistent memory, that it can recover from past errors by abstracting causal inductive rules about the world, and that CLIN is efficient in solving complex interactive tasks just using few trials achieving superior generalization performance.
>
> **> What are the design criteria and the rationale for the memory generator?**
>
> This is an important question! The rationale for the memory generator is to generate memories that **help the agent mentally simulate the causal effects of actions**, thus making more informed action choices in the future. This was missing in prior work, where memories were either task-specific factual instructions (e.g., "I should go to desk 1") or general hints ("Think about an object's usage when searching for it."). The idea of learning action causality has been previously explored in the formal planning literature (learning "action models") - our work is the first to take this idea into the modern context of LLMs, show its benefits, and make an important contribution.
>
> There are several ways this rationale can be operationalized, and in CLIN, we explore the simplest approach, namely guiding the memory generator to produce memories using two commonsense relations: "necessary" to indicate a positive effect of an action and "does not contribute" to indicate an undesired effect of an action. As you suggest, given this rationale, future work could explore alternative operationalizations of how action knowledge could be expressed, e.g., richer sentence templates. The important contribution of our work here is to introduce the general idea of learning causal action knowledge in a language-based setting to enable better performance in the future.
>
> **> How accurate are the generated memories?**
>
> Thank you for asking this question! While the final performance with memory is indicative of their effectiveness, we performed additional human evaluation of generated memory insights for correctness. For both the Gen-Env and Gen-Task setups, we randomly 10 task-environment combinations to evaluate the correctness of memories used in them, notably the meta-memory used for trial 0 (GEN) and memory adapted for the best trial (G+A). Two annotators rated all the insights (cohen's kappa 0.72) for correctness with reference to the respective gold trajectories, and the results are as follows:
>
> | | GEN-Env (Trial 0) | G+A (best trial) |
> |-----|-----|-----|
> | No. of insights | 100 | 105 |
> | Correct insights | 72.0% | **91.4%** |
> | Final score (on sampled tasks) | 39.1 | **55.9** |
>
>
> | | GEN-Task (Trial 0) | G+A (best trial) |
> |-----|-----|-----|
> | No. of insights | 98 | 107 |
> | Correct insights | 73.9% | **91.1%** |
> | Final score (on sampled tasks) | 43.7 | **58.1** |
>
> So, it is possible to have some meta-memories (here ~27%) that are not applicable for a new task or environments at Trial 0; however, with adaptation, in later trials, the correctness of the memory insights significantly improves (17-19% absolute), leading to a direct increase in task performance. We included this analysis in the Appendix.
>
> **> Does the performance improve because the LLM generates the correct memory, or just provides non-causal details of the task?**
>
> The memory generator is constrained (via prompting) to *always* produce causal statements, so there are no non-causal details in memory. From sample analysis, we observe that the generated memories are almost always correct with respect to the trials that have been seen, and empirically, performance improves using them (same task). However, when the environment or task changes, those learned memories may be over-general or incorrect (e.g., "activating the stove is necessary for heating water" is no longer correct in a new environment where the stove is broken). In such cases, we see an initial performance drop, but then further iterations cause those errors to be corrected, and performance improves even further. This can be seen in Figures 5a and 5b: In the same environment/task, CLIN improves (blue line), but then if the environment/task is changed, its performance drops as some learnings in memory are too specific (start of red line vs. end of blue line), but it then later recovers and improves even further (end of red line).

---

> ### Author Response · Authors · 2023-11-18
> **(Continued) Response to Reviewer 7v8j**
>
> **> The ablation doesn't show the usefulness of the causal abstraction**
>
> We apologize for being unclear here: When we drop the constraint that the memory should have a causal structure ("is necessary" / "does not contribute"), the memory generator can generate anything - and, in fact, stops generating causal abstractions and instead generates generic and often unhelpful advice (as others doing unconstrained memory generation have also found). Some example unconstrained generations are:
>
> 1. Avoid repeating actions: The agent performed the "look around" action twice at the beginning of the task. This is unnecessary and wastes time. Once you have a clear understanding of your surroundings, move on to the next step.
> 2. Be mindful of the order of actions: The agent moved to the workshop and read the instructions after looking around twice. It's better to first read the instructions to understand the task requirements.
> 3. Be clear with your actions: The agent made an ambiguous request when trying to mix the cup containing red paint. Be specific with your actions to avoid confusion.
> 4. Avoid unnecessary movements: The agent teleported to the art studio after picking up the paper. Plan your actions carefully to avoid unnecessary movements.
>
> The ablation (Figure 5d) thus shows the value of having a memory focused on **understanding the causal effects of actions** (rather than just any old reflection), following our rationale that this will help an agent mentally simulate their choices before acting. This is a key contribution of our work, and distinguishes it from other reflective approaches. For better readability, we included this example in the Appendix.
>
> **> What kind of information is still missing for the long tasks?**
>
> Longer tasks often require actions whose effects are not observed immediately but observed later in the time. Many longer tasks require frequent "waiting," allowing the environment to have the necessary changes (waiting for the bees to cross-pollinate). While our agent could not perform well at the initial trials later added a new memory: "Repeatedly waiting after opening the bee hive may be necessary to the growth of the cherry plant.". However, such a memory, while helpful, is often not precise about how long the agent needs to wait for the cross-pollination to happen, a stochastic process. Due to a lack of dense reward and sparse signal from the observations, our agent sometimes cannot complete longer tasks, which is indicated by Figure 4a.
>
> Figure 4c shows the % of episodes where there is a positive change in scores as compared to the Base agent (first trial). For shorter tasks, the room for improvement is less, as the average absolute score is higher as compared to longer tasks. However, we show higher relative improvements for longer tasks, indicating CLIN is able to bridge the performance gap by rapid adaptation.

---

> > ### Author Response · Authors · 2023-11-21
> > **Gentle reminder if there are any additional questions**
> >
> > Dear Reviewer,
> >
> > Thank you again for taking the time to review our paper. We hope we addressed all your concerns from the initial review. Since we are almost at the end of the discussion period (slightly more than a day left), please let us know if you have any remaining questions, and we will be happy to answer them as well.
> >
> > Best,
> > Authors

---

> > > ### Comment · Reviewer_7v8j · 2023-11-22
> > > **Thanks for the response!**
> > >
> > > I would like to thank the authors for providing additional discussion and evaluation of the accuracy of the generated memory. The response addressed many of my questions.
> > >
> > > My main question now is the generation and generalization of causal memory. The two templates that generate memory are "necessary" and "does not contribute". If we consider this in a task planning context, these two templates try to identify the preconditions of a task. This is a limited form of causal memory and depends on the environment setup. So a memory that is true for one environment may not be relevant at all in another environment. For example, whether "Going to the drawers may be necessary to find the remotecontrols" is useful depends on where the remote controls are; but "remotecontrols may be necessary to turn on the TV" can be applied across environments/tasks. The presented method does not differentiate between these types of memory. Without differentiating different types of causal memory, the generalization step presented is similar to providing an exploration bias, e.g. Fig 1 shows "moving to a different room may be necessary to find an object". This can be added by using only LLMs with a description of the environment without looking at the history of execution.

---

> ### Author Response · Authors · 2023-11-23
> **Thank you and follow-up discussion**
>
> Thanks for your response!
>
> We would like to just push back a little:
>
> **CLIN does differentiate between memories within a single task+environment (its normal memory) and when the task/environment changes (the meta-memory)**
>
> For example, within a single task+environment, it may learn "going to the kitchen is necessary for finding a cup". However, given a new task+environment, it will generate a meta-memory from previous memories of earlier (different) task+environments. CLIN is self-adaptive: the degree of generalization depends on the variation it's seen earlier. For example, if a cup has always been seen in the kitchen, it will retain "going to the kitchen is necessary for finding a cup." However if the cup appears in different places in different earlier trials, CLIN will drop or generalize (e.g., "going to a room is necessary for finding a cup") such a statement when building the initial meta-memory. In this way, CLIN automatically adapts to learn what is constant (e.g., the stove is always in the kitchen) and what is not.
>
> Similarly, in ALFWorld, if it has always seen remote controls to be found in drawer 1 or drawer 2, then the generalization would be “Going to the drawers may be necessary to find the remotecontrols.” While this looks simple for ALFWorld, in ScienceWorld, CLIN learns compositional generalization as a part of the meta memory:
>
> Task: boil chocolate.
>
> Environment 1: You start from art studio, Stove is working (revealed during interaction)
>
> Memory 1:
>
> Moving to the kitchen should be necessary to access cooking tools and ingredients. Turning on the stove should be necessary to boiling the chocolate.
>
> Task: boil chocolate.
>
> Environment 2: You start from kitchen, Stove is not working, lighter available (revealed during interaction)
>
> Memory 2:
>
> Using the lighter on the metal pot should be necessary to heat the chocolate. Turning on the stove does not contribute to boiling the chocolate if the stove is broken.
>
> Meta memory from Memory 1 and Memory 2:
>
> Moving to the kitchen should be necessary to access cooking tools and ingredients.
> Using a heat source (stove, lighter) on the container should be necessary to heat the substance.
> Turning on the stove does not contribute to the task if the stove is broken.
> CLIN can adapt further (for the specific task+environment): If such learnings in the meta-memory are violated in the new environment (e.g., although it’s always seen the cup in the kitchen, now it's suddenly no longer there!), then CLIN will continue to adapt by again generalizing or dropping such memories in the next iteration.
>
> In other words, the degree of specificity in the memory is constantly being updated, and the memory is being continually self-tuned to the tasks and environment. If the environment is not changing much, then memories will tend to be (desirably) fairly specific. If the environment is changing a lot, then memories will be more general. This self-regulation of CLIN is an important point and a key contribution of the work that distinguishes it from earlier work. We hope this addresses your concern here. We will make this clearer in the paper!

---

> ### Author Response · Authors · 2023-11-23
> **(Continued) Follow-up discussion**
>
> **Just looking at the task description and the initial state of the environment is not enough to produce grounded causal rules**
>
> Meta-memory when only the task description and the initial state are provided (Note ScienceWorld is a partially observable world)
>
> NEW TASK: Your task is to grow an orange. This will require growing several plants and then being crosspollinated to produce fruit.  Seeds can be found in the bathroom. To complete the task, focus on the grown orange.
> This room is called the bedroom. In it, you see: the agent, a substance called air, a bed. On the bed is: a mattress. On the mattress is: a white pillow, a book shelf (containing A book (Moby Dick) titled Moby Dick by Herman Melville, A book (Pride and Prejudice) titled Pride and Prejudice by Jane Austen, A book (Sherlock Holmes) titled Sherlock Holmes by Arthur Conan Doyle), a closet: The closet door is closed, a drawing, a table: On the table is: nothing. You also see: A door to the hallway (that is open).
>
> 1. Air may be necessary to grow an orange.
> 2. The bed does not contribute to growing an orange.
> 3. The mattress does not contribute to growing an orange.
> 4. The white pillow does not contribute to growing an orange.
> 5. The book shelf may not contribute to growing an orange.
> 6. The books on the shelf do not contribute to growing an orange.
> 7. The closet may be necessary to store tools or materials for growing an orange.
> 8. The drawing does not contribute to growing an orange.
> 9. The table may be necessary to place the plant pots for growing an orange.
> 10. The door to the hallway may be necessary to move between rooms and gather necessary items for growing an orange.
>
> Without any interaction with the world, it could only generate trivial insights like the above that describe whether each object it observes at the start of the trial "is necessary/ does not contribute" to the task. Unless the agent navigates to an appropriate room, it won't see objects like "seed jar", "flower pot", "jug", "bee hive" etc. that are essential to complete the task.
> On the other hand, CLIN's meta-memory is composed of much more informative insights based on interactions with the environment from its previous experience on growing orange in **different** environment configurations. The learnings below mention essential objects, relevant actions, and how they "are necessary / do not contribute" to the task and subgoals within the task.
>
> 1. Looking around may be necessary to identify the objects in the room.
> 2. Moving to the appropriate room should be necessary to find the seed jar.
> 3. Picking up the seed jar should be necessary to start the process of growing an orange.
> 4. Moving to the greenhouse should be necessary to plant the seeds.
> 5. Placing the seeds in the flower pot should be necessary to plant the seeds.
> 6. Picking up the jug may be necessary to water the seeds.
> 7. Filling the jug at the sink should be necessary to get water for the seeds before pouring it into the flower pot.
> 8. Pouring water from the jug into the flower pot should be necessary to water the seeds.
> 9. Waiting may be necessary for the seeds to grow into an orange plant.
> 10. Repeatedly waiting may contribute to the growth of the orange plant.
> 11. Opening the bee hive may be necessary to crosspollinate the plants.
> 12. Repeatedly waiting after opening the bee hive may contribute to the growth of the orange plant.
>
> CLIN produces generalized exploration bias, self-adapt if necessary, the novelty of our framework. When there is a huge search space for possible actions, any attempt to focus on desirable (part of the) trajectories (or memories) to glean relevant insights can be seen as a necessary exploration bias to perform well in unseen tasks/environments.

---

### Official Review · Reviewer_FEc9 · 2023-10-31

**Soundness:** 3 good
**Presentation:** 3 good
**Contribution:** 2 fair
**Rating:** 6
**Confidence:** 4

**Summary:**

This paper studies the continual learning abilities of LLM agents and proposes CLIN that maintains a memory of causal abstractions to help in future trials by reflecting on the last trial and current memory.

**Strengths:**

1. The paper is clearly written and easy to follow.
2. The experiments demonstrate a large improvement in multiple environments.
3. The illustrations in the paper are informative.

**Weaknesses:**

1. The novelty is somewhat limited. Specifically, the authors discussed the difference between their method and Reflexion. But it is not clear if the causal abstractions can be generalized in distinct environments with different goals. Even if they can, I assume that the summaries kept in memory are pretty high-level and abstract. Can these summaries still benefit the performance of individual tasks? The experiments in the ScienceWorld benchmark make sense as different tasks share common rules. More experiments would be good to support the method, such as the benchmarks used in Reflexion (e.g., ALFWorld that the authors used as examples), where common rules may be hard to find useful for individual tasks.
2. More comparisons to related works can be added, such as [1, 2] which also improve iteratively via trials.

[1] AdaPlanner: Adaptive Planning from Feedback with Language Models. Sun et al.
[2] Reason for Future, Act for Now: A Principled Framework for Autonomous LLM Agents with Provable Sample Efficiency. Liu et al.

**Questions:**

See weaknesses.

---

> ### Author Response · Authors · 2023-11-18
> **Response to Reviewer FEc9**
>
> Thank you for taking the time to review our paper! We were happy to read that you appreciated our main points: that our framework outperforms many existing works by a large margin and that our work is informative and well-presented.
>
> **> Can you clarify the novelty?**
>
> There are two key novelties:
>
> 1. While the idea of having a memory is not new, our novel contribution is to make this a **dynamic, evolving memory over time**, in contrast to the short-term "reflect, use, then discard" approach used in Reflexion and other memory-based agents. This allows CLIN to progressively build insights about the world over multiple experiences in different environments. It also allows CLIN to recover from bad or not-useful learnings in memory, as these will get dropped when they are found not to help as the memory is updated.
>
> 2. The **role** of the memory is itself novel, namely to help the agent mentally simulate the (learned) causal effects of actions, thus making more informed action choices in the future. This was missing in prior work, where memories were either task-specific factual instructions (e.g., "I should go to desk 1") or general hints ("Think about an object's usage when searching for it."). The idea of learning the causal effects of actions ("action models"), to help simulation/planning, has been previously explored in formal planning. Our work is the first to take this idea into the modern context of LLMs, and show its benefits.
>
> **> Can causal abstractions really be generated from different environments or tasks, and if so, do they still provide benefit?**
>
> Yes, such generalized causal abstractions can indeed be generated: Our meta-memory (abstracted over past trials in different tasks or environments) contains such abstractions. For example, from the causal rules (1) "Using the lighter on the metal pot should be necessary to heat the water" and (2) "Turning on the stove should be necessary to create heat", CLIN generates the abstract rule "Using a heat source (stove, lighter) on the container should be necessary to heat a substance" (see Figure 3). Empirically, we do see that meta-memory provides benefits in novel tasks, both immediately and after further learning (Figure 5a and 5b, the with-meta-memory red line is above the blue line).
>
> **> More support showing CLIN can generate generalized insights where common rules may be hard to find useful for individual tasks such as ALFWorld.**
>
> Indeed, tasks in ALFWorld require specific insights such as "going to fridge 1 is necessary to find the tomato." However, in terms of complexity and task type, the Pick&Place tasks in ScienceWorld subsume most of the ALFWorld tasks, and agents (e.g., ReAct) that perform well on ALFWorld perform considerably poorly in similar tasks in ScienceWorld. On the contrary, CLIN achieves perfect scores in such tasks during generalization.
>
> As a prototype for generalized memory in ALFWorld, we accrue two successful trajectories of Pick&Place tasks, and generate a meta-memory using CLIN's causal abstraction for a target task: "Find two remotecontrols and put them in armchair":
>
> 1. Going to the drawers may be necessary to find the remotecontrols.
> 2. Opening the drawers should be necessary to access the remotecontrols.
> 3. Taking the remote controls from the drawer should be necessary to obtain the remotecontrols.
> 4. Going to the armchair 1 should be necessary to place the remotecontrols.
> 5. Putting remotecontorls in/on armchair 1 should be necessary to complete the task.
>
> We manually compared the correctness of such generalized insights with the underlying PDDL file that spawns the task and found that the memory captures most of the necessary (causal) information needed to solve the target task, indicating CLIN-style meta-memory will be effective in ALFWorld-like environments too.
>
> **> Related work: AdaPlanner and "Reason for the Future, Act Now"**
>
> Thank you for pointing out these interesting works, we will add them! As points of contrast, AdaPlanner's primary strength is adaptation/self-refinement of a plan during execution, rather than continual learning between plans/tasks. It does have a skill memory of successful plans, but does not abstract over them nor "learn" from failures beyond a datapoint. "Reason for Future, Act for Now", a concurrent work, is also interesting: they propose a memory-based approach to algorithmically reason over possible hypotheses for the next goal and action over a longer trajectory. However, they also do not persist insights in one trajectory to generalize over unseen tasks or environment – a key difference between recent works and our work.
>
> As a prototype, we implemented a version of AdaPlanner for Pick&Place tasks in ScienceWorld, similar to those it has been predominantly successful in ALFWorld. On such tasks, AdaPlanner achieves an average score of 38.3, slightly lower than ReFlexion (40.7), whereas CLIN (our agent) achieves a perfect score of 100 in all such (Pick&Place) tasks.

---

> > ### Author Response · Authors · 2023-11-21
> > **Gentle reminder if there are any additional questions**
> >
> > Dear Reviewer,
> >
> > Thank you again for taking the time to review our paper. We hope we addressed all your concerns from the initial review. Since we are almost at the end of the discussion period (slightly more than a day left), please let us know if you have any remaining questions, and we will be happy to answer them as well.
> >
> > Best,
> > Authors

---

> > > ### Comment · Reviewer_FEc9 · 2023-11-22
> > >
> > > Thank the authors for the clarification! After reading the response and other reviewers' comments, I feel that the method is not general enough. For example, the memory only contains relations with a fixed and predefined format, e.g., "x may be necessary to y", "x may be contribute to y", which is highly dependent on the task environment. It's not clear if it is easy to extract the necessary learnings in more abstract tasks, such as interactive coding tasks. Besides, in the ALFWorld experiment in the rebuttal,  I am not sure if the learned memory is general enough, e.g., "Opening the drawers should be necessary to access the remotecontrols" might not be true in a new task that has the remotecontrols at other places. For these reasons, I decided to keep my original score.

---

> > > > ### Author Response · Authors · 2023-11-23
> > > > **Thank you and follow-up discussion**
> > > >
> > > > Thanks for your response!
> > > >
> > > > We would like to just push back a little:
> > > >
> > > > **CLIN does differentiate between memories within a single task+environment (its normal memory) and when the task/environment changes (the meta-memory)**
> > > >
> > > > For example, within a single task+environment, it may learn "going to the kitchen is necessary for finding a cup". However, given a new task+environment, it will generate a meta-memory from previous memories of earlier (different) task+environments. CLIN is self-adaptive: the degree of generalization depends on the variation it's seen earlier. For example, if a cup has always been seen in the kitchen, it will retain "going to the kitchen is necessary for finding a cup." However if the cup appears in different places in different earlier trials, CLIN will drop or generalize (e.g., "going to a room is necessary for finding a cup") such a statement when building the initial meta-memory. In this way, CLIN automatically adapts to learn what is constant (e.g., the stove is always in the kitchen) and what is not.
> > > >
> > > > Similarly, in ALFWorld, if it has always seen remote controls to be found in drawer 1 or drawer 2, then the generalization would be “Going to the *drawers* may be necessary to find the remotecontrols.” While this looks simple for ALFWorld, in ScienceWorld, CLIN learns compositional generalization as a part of the meta memory:
> > > >
> > > > Task: boil chocolate.
> > > >
> > > > Environment 1: You start from art studio, Stove is working (revealed during interaction)
> > > >
> > > > Memory 1:
> > > >
> > > > Moving to the kitchen should be necessary to access cooking tools and ingredients.
> > > > Turning on the stove should be necessary to boiling the chocolate.
> > > >
> > > > Task: boil chocolate.
> > > >
> > > > Environment 2: You start from kitchen, Stove is not working, lighter available (revealed during interaction)
> > > >
> > > > Memory 2:
> > > >
> > > > Using the lighter on the metal pot should be necessary to heat the chocolate.
> > > > Turning on the stove does not contribute to boiling the chocolate if the stove is broken.
> > > >
> > > > Meta memory from Memory 1 and Memory 2:
> > > >
> > > > 1. Moving to the kitchen should be necessary to access cooking tools and ingredients.
> > > > 2. Using a heat source (stove, lighter) on the container should be necessary to heat the substance.
> > > > 3. Turning on the stove does not contribute to the task if the stove is broken.
> > > >
> > > > CLIN can adapt further (for the specific task+environment): If such learnings in the meta-memory are violated in the new environment (e.g., although it’s always seen the cup in the kitchen, now it's suddenly no longer there!), then CLIN will continue to adapt by again generalizing or dropping such memories in the next iteration.
> > > >
> > > > In other words, the degree of specificity in the memory is constantly being updated, and the memory is being continually self-tuned to the tasks and environment. If the environment is not changing much, then memories will tend to be (desirably) fairly specific. If the environment is changing a lot, then memories will be more general. This self-regulation of CLIN is an important point and a key contribution of the work that distinguishes it from earlier work. We hope this addresses your concern here. We will make this clearer in the paper!

---

> ### Author Response · Authors · 2023-11-23
> **(Continued) Follow-up discussion**
>
> **Just looking at the task description and the initial state of the environment is not enough to produce grounded causal rules**
>
> Meta-memory when only the task description and the initial state are provided (Note ScienceWorld is a partially observable world)
>
> NEW TASK: Your task is to grow an orange. This will require growing several plants and then being crosspollinated to produce fruit.  Seeds can be found in the bathroom. To complete the task, focus on the grown orange.
> This room is called the bedroom. In it, you see: the agent, a substance called air, a bed. On the bed is: a mattress. On the mattress is: a white pillow, a book shelf (containing A book (Moby Dick) titled Moby Dick by Herman Melville, A book (Pride and Prejudice) titled Pride and Prejudice by Jane Austen, A book (Sherlock Holmes) titled Sherlock Holmes by Arthur Conan Doyle), a closet: The closet door is closed, a drawing, a table: On the table is: nothing. You also see: A door to the hallway (that is open).
>
> 1. Air may be necessary to grow an orange.
> 2. The bed does not contribute to growing an orange.
> 3. The mattress does not contribute to growing an orange.
> 4. The white pillow does not contribute to growing an orange.
> 5. The book shelf may not contribute to growing an orange.
> 6. The books on the shelf do not contribute to growing an orange.
> 7. The closet may be necessary to store tools or materials for growing an orange.
> 8. The drawing does not contribute to growing an orange.
> 9. The table may be necessary to place the plant pots for growing an orange.
> 10. The door to the hallway may be necessary to move between rooms and gather necessary items for growing an orange.
>
> Without any interaction with the world, it could only generate trivial insights like the above that describe whether each object it observes at the start of the trial "is necessary/ does not contribute" to the task. Unless the agent navigates to an appropriate room, it won't see objects like "seed jar", "flower pot", "jug", "bee hive" etc. that are essential to complete the task.
> On the other hand, CLIN's meta-memory is composed of much more informative insights based on interactions with the environment from its previous experience on growing orange in **different** environment configurations. The learnings below mention essential objects, relevant actions, and how they "are necessary / do not contribute" to the task and subgoals within the task.
>
> 1. Looking around may be necessary to identify the objects in the room.
> 2. Moving to the appropriate room should be necessary to find the seed jar.
> 3. Picking up the seed jar should be necessary to start the process of growing an orange.
> 4. Moving to the greenhouse should be necessary to plant the seeds.
> 5. Placing the seeds in the flower pot should be necessary to plant the seeds.
> 6. Picking up the jug may be necessary to water the seeds.
> 7. Filling the jug at the sink should be necessary to get water for the seeds before pouring it into the flower pot.
> 8. Pouring water from the jug into the flower pot should be necessary to water the seeds.
> 9. Waiting may be necessary for the seeds to grow into an orange plant.
> 10. Repeatedly waiting may contribute to the growth of the orange plant.
> 11. Opening the bee hive may be necessary to crosspollinate the plants.
> 12. Repeatedly waiting after opening the bee hive may contribute to the growth of the orange plant.
>
> CLIN produces generalized exploration bias, self-adapt if necessary, the novelty of our framework. When there is a huge search space for possible actions, any attempt to focus on desirable (part of the) trajectories (or memories) to glean relevant insights can be seen as a necessary exploration bias to perform well in unseen tasks/environments.

---

### Official Review · Reviewer_9kZa · 2023-10-31

**Soundness:** 3 good
**Presentation:** 3 good
**Contribution:** 2 fair
**Rating:** 5
**Confidence:** 4

**Summary:**

This paper presents a continuously learning language agent called CLIN, which improves the language-based agents. Unlike previous models, CLIN improves its performance without the need for parameter updates. CLIN uses a system centered on causal abstractions and a dynamic textual memory that regularly updates after each trial, the agent gradually learns and applies new knowledge for future trials. This framework is evaluated in the ScienceWorld benchmark, demonstrating its ability to adapt across different tasks and environments.

**Strengths:**

- The underlying mechanism of using causal abstractions for memory storage demonstrates a novel learning methodology.
- The experiments conducted using the ScienceWorld benchmark are comprehensive and rigorous.
- The paper is well-structured, making it easy to understand.

**Weaknesses:**

- The CLIN system is built on past experiences carrying forward context to new experiences. When there are potential issues such as misleading context, it will affect its performance.
- Although CLIN claims to have the ability of self-exploration, it primarily focuses on known information, which may affect its performance in new environments.
- Although CLIN presents some potential improvements over existing methods, it still does not have significant breakthroughs within the framework of current works.

**Questions:**

- CLIN's textual memory seems to depend heavily on causal abstractions. How does the system handle tasks where the relations are not causal or obscure?
- How robust is the system when facing incorrect or misleading information stored in its memory? Is there any mechanism in place to correct or override these inaccuracies?
- How does the system handle potential changes in state that are not immediately reacted to an action?
- Could you provide more insights on how the executor modifies an action when it doesn't align with valid actions? Specifically, how does it ensure that the modified action still aligns with the intended goal?

---

> ### Author Response · Authors · 2023-11-18
> **Response to Reviewer 9kZa**
>
> Thank you for taking the time to review our paper! We were happy to read that you appreciated our main points: that our work demonstrates a novel learning methodology with continual learning and causal memory, that our experiments are extensive and rigorous, and that our work is well-presented.
>
> **> CLIN carries forward context to new experiences. Will misleading context affect performance?**
>
> In the short term, yes: errors in the memory will affect performance. However, note that CLIN also recovers from such errors, as the memory is updated after each trial, thus evolving over time - this dynamic, evolving memory is a key contribution of the work. Specifically, if a causal rule is wrong or unhelpful in a given trial, then CLIN will drop or update it during the memory update process, after it is observed to be ineffective. We see exactly this phenomenon in Figures 5a and 5b: In the same environment/task, CLIN improves (blue line), but then if the environment/task is changed, its performance drops as some learnings in memory are too specific (start of red line vs. end of blue line), but it then later recovers and improves even further (end of red line).
>
> **> Is CLIN's ability to self-explore limited by known information?**
>
> CLIN's memory both guides CLIN towards a solution (via the "is necessary for" links), but also explicitly encourages exploration if earlier attempts fail (via the "does not contribute to" links). For example, if in trial T, action X did not contribute to goal Y, this will be noted in memory ("X does not contribute to Y"), causing CLIN to explore a different action X' in the next trial. In this way, CLIN maintains a balance between exploration and exploitation, depending on the success and failures in earlier trials.
>
> **> What are the significant breakthroughs in CLIN?**
>
> There are two significant advances in CLIN:
>
> 1. While the idea of having a memory is not new, our novel contribution is to make this a **dynamic, evolving memory over time**, in contrast to the short-term "reflect, use, then discard" approach used in Reflexion and other memory-based agents. This allows CLIN to progressively build insights about the world over multiple experiences in different environments. It also allows CLIN to recover from bad or not-useful learnings in memory, as these will get dropped when they are found not to help as the memory is updated.
> 2. The **role** of the memory is itself novel, namely to help the agent mentally simulate the (learned) causal effects of actions, thus making more informed action choices in the future. This was missing in prior work, where memories were either task-specific factual instructions (e.g., "I should go to desk 1") or general hints ("Think about an object's usage when searching for it."). The idea of learning the causal effects of actions ("action models"), to help simulation/planning, has been previously explored in formal planning. Our work is the first to take this idea into the modern context of LLMs, and show its benefits.
>
> We have updated the Introduction in the revised paper to clarify these better.
>
> **> How does the system handle tasks where the relations are not causal or obscure?**
>
> In general, non-causal knowledge has a causal import, which we observe is then captured in the memory. For example, "the block is in the workshop" would be captured by the causal expression "moving to the workshop may be necessary to find the block." In general, causal abstractions indirectly capture factual knowledge relevant to the tasks being considered.
>
> There are definitely other kinds of relations that are required for ScienceWorld -- like retrieving property knowledge. An agent needs to know the melting points of objects, the needs of living things, the affordances of objects, etc., which are critical to solving the tasks. We empirically show that our agents are achieving perfect scores (add numbers) in many of these tasks requiring apparent non-causal knowledge. However, qualitative analysis reveals that CLIN is largely able to convert such knowledge with the structured causal import and effectively use it for completion of the tasks.
>
> **> How robust is the system when facing incorrect or misleading information stored in its memory? Is there any mechanism in place to correct or override these inaccuracies?**
>
> Yes - in fact, a key contribution of CLIN is its ability to correct errors in its memory via its dynamic evolution over time, unlike prior work. The memory is always updated after each trial. If the latest trace indicates some memories are unhelpful or simply wrong, they will not be retained in the updated memory, given the LM is following its prompt-based instructions correctly. Please see our **new results** in the "[General Response to Reviewers](https://openreview.net/forum?id=d5DGVHMdsC&noteId=YGdT6et2uW)" (also Appendix C, Table 2 in the revised paper) on CLIN's recovery from wrong knowledge in its memory via continual learning.

---

> ### Author Response · Authors · 2023-11-18
> **(Continued) Response to Reviewer 9kZa**
>
> **> How does the system handle potential changes in state that are not immediately reacted to an action?**
>
> These do indeed occur in ScienceWorld, e.g., water changes to vapor multiple several steps after the action "activate stove" with a pot of water on it. In this case, the actions "wait" or "take the water temperature" may be executed. CLIN learns through experimentation that some tasks require such actions, and such learnings may then be added to memory (e.g., when growing plants, "waiting may be necessary to allow the plants to grow", Figure 10).
>
> **> How does the executor modify an action when it doesn't align with valid actions?**
>
> If the executor generates an invalid (unexecutable) action, a simple check detects this and the executor is iteratively called again (with a note not to generate the invalid action) until a valid action is produced (or a max number of attempts is exceeded). While refinement, the executor is still conditioned on the intended goal provided by the controller. This utility essentially implements a self-refine technique [Madan et al., 2023](https://arxiv.org/pdf/2303.17651.pdf) to address this problem in the executor module.

---

> > ### Author Response · Authors · 2023-11-21
> > **Gentle reminder if there are any additional questions**
> >
> > Dear Reviewer,
> >
> > Thank you again for taking the time to review our paper. We hope we addressed all your concerns from the initial review. Since we are almost at the end of the discussion period (slightly more than a day left), please let us know if you have any remaining questions, and we will be happy to answer them as well.
> >
> > Best,
> > Authors

---

> > > ### Comment · Reviewer_9kZa · 2023-11-22
> > >
> > > Thanks for the author's response. Most of my concerns have been addressed, for now I will maintain my current score and continue to pay attention to other reviews and discussions.

---

### Official Review · Reviewer_dtLL · 2023-10-31

**Soundness:** 3 good
**Presentation:** 3 good
**Contribution:** 2 fair
**Rating:** 5
**Confidence:** 3

**Summary:**

This paper proposes to use a memory module to help LLM-based agents to do continual learning using interactions with the world. The memory module takes an novel form of causal abstractions to summarize the agent's interaction trace into knowledge that can be reused for future interaction.

**Strengths:**

Significance: While the idea of having a memory to store experiences is not new, the novelty of the work lies in using LLM to summarize the experience into some form of knowledge, here in causal abstractions. I think this work could open door to more advanced and systematic study of how LLM agents can learn, i.e. gain knowledge from raw information, based on online interactions with the world.

For example, in a new gaming environment with different physics laws, the LLM-based agent needs to interact with the world to gain information on the actual physics in the new environment. There needs to be a process of learning and obtaining abstractions from these experiences to get generalizable laws. This can potentially help LLM gain new knowledge or domain-specific knowledge about unseen world and unseen domains.

Experimental Results: the experimental results and comparison with baselines are convincing.

**Weaknesses:**

1. Technical Novelty: the idea of having a memory of past experiences and learned summary of knowledge is not very new; while this work proposes that the content in the memory is important, i.e., using causal abstractions helps agent's learning, the method is not that different in terms of the general idea.

I think the causal abstraction seems to be only a specific instantiation of imposing some prior structure of memory. The authors would need to give more evidence on why this is universally applicable.

2. The memory entirely depends on the agent's own exploration trace, so if the agent cannot explore enough and find some useful information, the agent cannot gain the corresponding knowledge.

3. I think usually for an agent to be "continually learning", it needs to adapt to many different tasks over time. However, this work focus on accumulating knowledge continually for one task, which is not really continual adaptation.

**Questions:**

1. When generating causal abstractions, how does the LLM resolves credit assignment problem? If the agent do several actions and results in several different results, how does it know what actions contribute (and not contribute) to what result?

2. Curious about whether the causal abstractions generation depends on LLM' known commonsense knowledge about the world, or depends on its own reasoning ability. What happens if the world follows some new logic, and does the causal abstraction generation still be correct?

3. If the environment dynamics changes and old causal relationships are wrong, does the agent know when to update and unlearn the original knowledge?

---

> ### Author Response · Authors · 2023-11-18
> **Response to Reviewer dtLL**
>
> Thank you for taking the time to review our paper! We were happy to read that you appreciated our main points: that CLIN can continually gain new knowledge based on online interactions with the world, that CLIN can abstract the world model (actual physics) of a new environment, and that our experiments are sound.
>
>
> **> How novel is CLIN? (the idea of having a memory of past experiences is not new)**
>
> There are two key novelties:
>
> 1. Indeed, the idea of having a memory is not new. Our novel contribution is to make this a **dynamic, evolving memory over time**, in contrast to the short-term "reflect, use, then discard" approach used in Reflexion and other memory-based agents. This allows CLIN to progressively build insights about the world over multiple experiences in different environments. It also allows CLIN to recover from bad or not-useful learnings in memory, as these will get dropped when they are found not to help as the memory is updated.
>
> 2. The **role** of the memory is itself novel, namely to help the agent mentally simulate the (learned) causal effects of actions, thus making more informed action choices in the future. This was missing in prior work, where memories were either task-specific factual instructions (e.g., "I should go to desk 1") or general hints ("Think about an object's usage when searching for it."). The idea of learning the causal effects of actions ("action models"), to help simulation/planning, has been previously explored in formal planning. Our work is the first to take this idea into the modern context of LLMs, and show its benefits.
>
> We will make this clearer in newer versions of the paper and have updated the Introduction to clarify these better.
>
>
> **> Does the memory depend entirely on the agent's exploration trace? If the agent doesn't explore enough, can the agent gain the required knowledge?**
>
> Like any agent, the agent learns from exploration, so some required knowledge (e.g., a fridge is in the kitchen) necessarily needs exploration. If CLIN doesn't explore enough (e.g., it fails to find a fridge in the living room), it can gain the required knowledge in the next iteration by adding a memory: "going to the living room does not contribute to finding a fridge". This "negative knowledge" biases CLIN to explore in a different direction in the future, helping it gain the required knowledge for the task, and balance exploration and exploitation.
>
>
> **> Is CLIN just continually learning for a single task?**
>
> No, we applied and evaluated it in two settings: (a) single task ("adapt" setting), (b) multiple different environments (Gen-Env), and tasks (Gen-Task) ("generalization" setting). In the multiple-task setting, given a new (previously unseen) task, the meta-memory is first built from traces on different training tasks, and then applied to the new, unseen task (see Gen-Task in Section 4, and Figure 5c). For example, CLIN performs a handful of boiling tasks before solving an unseen freezing task. The fact it can perform well on unseen tasks is important and an important contribution - we will highlight this more in the paper.
>
>
> **> When generating causal abstractions, how does the LLM resolve the credit assignment problem?**
>
> Good question: By informed trial and error, using commonsense. When initially generating causal abstractions, the LLM is biased towards proposing commonsense abstractions vs. nonsensical abstractions (e.g., it is more likely to propose "opening the kitchen door may be necessary to move to the kitchen" than "looking around may be necessary to move to the kitchen"). Second, when the memory is updated, regenerating a new set of causal abstractions based on the old ones and new experiences, we rely on the LLM to follow its prompt-based instructions correctly, in particular, dropping memories that are unhelpful or simply wrong. [Madan et al., 2023](https://arxiv.org/pdf/2303.17651.pdf]) show LLMs are good at refining their outputs, and [Ma et al., 2023](https://eureka-research.github.io/assets/eureka_paper.pdf) show that LLMs are powerful reward generators. Mirroring these two insights, we find similar phenomena: for example, our LLM-based memory generator can connect an action (activate stove) and an observation (stove is broken) using its prior commonsense to propose a rule that activating stove when it is broken does not contribute to boiling the water. Of course, such proposals may sometimes be wrong, in which case they should later be dropped in subsequent trials when they hurt or are unhelpful. Please see **our new results** in the "[General Response to Reviewers](https://openreview.net/forum?id=d5DGVHMdsC&noteId=YGdT6et2uW)" (also Appendix C, Table 2 in the revised paper) on evidence that CLIN is able to recover from wrong knowledge in its memory via continual learning.

---

> ### Author Response · Authors · 2023-11-18
> **(Continued) Response to Reviewer dtLL**
>
> **> What happens if the world follows some new logic, and does the causal abstraction generation still be correct?**
>
> This would be problematic! CLIN avoids thousands of trials (e.g., in RL) by leveraging the LLM's commonsense, so in a "nonsensical" world, this would not be effective. In fact, this is the key "breakthrough" insight of language-based agents: that with modern LLMs, commonsense can replace blind trial-and-error in real-world-like environments.
>
> **> If the environment dynamics change and old causal relationships are wrong, does the agent know when to update and unlearn the original knowledge?**
>
> Yes - the memory is updated after each trial, and as causal insights learned from past experiences no longer apply, or are found now to be incorrect, the LM will adapt or drop them based on the information provided in the latest trial. For example, CLIN learned that "activating the stove is necessary to boil a substance" but then encountered a trial with a broken stove. It subsequently updated that rule in the memory update to "activating the stove does not contribute to boiling a substance when the stove is broken." Again, this update process is not perfect, but our experiments show the memory dynamically evolves to a high-performing result, an exciting and novel finding.

---

> > ### Author Response · Authors · 2023-11-21
> > **Gentle reminder if there are any additional questions**
> >
> > Dear Reviewer,
> >
> > Thank you again for taking the time to review our paper. We hope we addressed all your concerns from the initial review. Since we are almost at the end of the discussion period (slightly more than a day left), please let us know if you have any remaining questions, and we will be happy to answer them as well.
> >
> > Best,
> > Authors

---

> > > ### Comment · Reviewer_dtLL · 2023-11-21
> > > **Thank you for your response**
> > >
> > > Thank you the authors for your response! I acknowledge that the comments are received and I appreciate the further discussion.

---

### Author Response · Authors · 2023-11-18
**General Response to Reviewers**

We would like to thank all the reviewers for their valuable feedback. We are encouraged that they find our approach novel and can inspire more works on continually learning language agents (Reviewers `dtLL`, `9kZa`), efficient in rapid adaptation and generalization (Reviewers `7v8j`, `FEc9`), with comprehensive results surpassing many baselines (Reviewers `FEc9`, `7v8j`), backed by sound and rigorous experiments (Reviewers `dtLL`, `9kZa`). We have addressed reviewers' comments individually and look forward to a fruitful interaction during the author response period.

Notably, we updated our manuscript by adding the **new results** for memory correctness, clarified the narrative for the ablation study, and updated the Introduction to clarify our novelty and contributions:

**More evidence for memory correctness/error recovery via continual learning (updated in the Appendix C, Table 2):**
For both the Gen-Env and Gen-Task setups, we randomly 10 task-environment combinations to evaluate the correctness of memories used in them, notably the meta-memory used for trial 0 (GEN) and memory adapted for the best trial (G+A). Two annotators rated all the insights (cohen's kappa 0.72) for correctness with reference to the respective gold trajectories, and the results are as follows:
| | GEN-Env (Trial 0) | G+A (best trial) |
|-----|-----|-----|
| No. of insights | 100 | 105 |
| Correct insights | 72.0% | **91.4%** |
| Final score (on sampled tasks) | 39.1 | **55.9** |

| | GEN-Task (Trial 0) | G+A (best trial) |
|-----|-----|-----|
| No. of insights | 98 | 107 |
| Correct insights | 73.9% | **91.1%** |
| Final score (on sampled tasks) | 43.7 | **58.1** |

**Clarified novelty and contributions (updated in the Introduction)**:
1. While the idea of having a memory is not new, our novel contribution is to make this a **dynamic, evolving memory over time**, in contrast to the short-term "reflect, use, then discard" approach used in Reflexion and other memory-based agents. This is allows CLIN to progressively build insights about the world over multiple experiences in different environments. It also allows CLIN to recover from bad or not-useful learnings in memory, as these will get dropped when they are found not to help as the memory is updated.

2. The **role** of the memory is itself novel, namely to help the agent mentally simulate the (learned) causal effects of actions, thus make more informed action choices in future. This was missing in prior work, where memories were either task-specific factual instructions (e.g., "I should go to desk 1") or general hints ("Think about an object's usage when searching for it."). The idea of learning the causal effects of actions ("action models"), to help simulation/planning, has been previous explored in formal planning. Our work is the first to take this idea into the modern context of LLMs, and show its benefits.


Here, we provide a [video](https://drive.google.com/file/d/1e18KvMytNq5xU0cJOxjmIPTxD_ASRErv/view?usp=drive_link) showing our agent performing actions in ScienceWorld, creating casual abstractions as memory, and finally solving the task after few unsuccessful trials using continually evolving memory. Note that ScienceWorld is visually represented here for better showcasing—the (original) simulation (used in the paper) is text-only.

---

### Meta-Review · Area_Chair_b6Ke · 2023-12-06

**Metareview:**

Concerns regarding the specificity of the approach to environment/benchmark used were repeatedly raised. The authors tried to address them via prototype experiments in ALFWorld. This seems to expand the paper in the right direction, but require work that is more methodological and deep than is possible within a response period. As such, the reviewers were mostly not convinced given the examples traces from ALFWorld, although they appreciate the effort the authors put into getting something for response in such a short time.

**Justification For Why Not Higher Score:**

See above

**Justification For Why Not Lower Score:**

N/A

---

### Decision · Program_Chairs · 2024-01-16

Reject